# GRAIN: Exact Graph Reconstruction from Gradients

**Maria Drencheva**[1], **Ivo Petrov**[1], **Maximilian Baader**[2], **Dimitar I. Dimitrov**[1,2], **Martin Vechev**[1,2]

[1] INSAIT, Sofia University "St. Kliment Ohridski"   [2] ETH Zurich

{maria.drencheva, ivo.petrov, dimitar.iliev.dimitrov}@insait.ai [1]
{mbaader,martin.vechev}@inf.ethz.ch [2]

## Abstract

Federated learning claims to enable collaborative model training among multiple clients with data privacy by transmitting gradient updates instead of the actual client data. However, recent studies have shown the client privacy is still at risk due to the, so called, gradient inversion attacks which can precisely reconstruct clients' text and image data from the shared gradient updates. While these attacks demonstrate severe privacy risks for certain domains and architectures, the vulnerability of other commonly-used data types, such as graph-structured data, remain under-explored. To bridge this gap, we present GRAIN, the first exact gradient inversion attack on graph data in the honest-but-curious setting that recovers both the structure of the graph and the associated node features. Concretely, we focus on Graph Convolutional Networks (GCN) and Graph Attention Networks (GAT) – two of the most widely used frameworks for learning on graphs. Our method first utilizes the low-rank structure of GNN gradients to efficiently reconstruct and filter the client subgraphs which are then joined to complete the input graph. We evaluate our approach on molecular, citation, and social network datasets using our novel metric. We show that GRAIN reconstructs up to $80\%$ of all graphs exactly, significantly outperforming the baseline, which achieves up to 20% correctly positioned nodes.

## 1 Introduction

Graph Neural Networks (GNNs) (Scarselli et al., 2009) have shown a great promise in learning on graph-structured data like social networks, traffic flows, molecules, as well as healthcare and income data. Many of these applications, however, require large quantities of private data, which can be hard to collect due to privacy regulations and the reluctance of users to share their data due to fear of losing competitive advantage. This has naturally led to widespread use of GCNs and GATs alongside Federated Learning (FL) which promises to protect the sensitive data of users (Xie et al., 2021; Zhang et al., 2021; Zhu et al., 2022; Lee et al., 2022; Lou et al., 2021; Peng et al., 2022).

However, the privacy of client data in FL in different domains including images (Zhang et al., 2023), text (Petrov et al., 2024), and tabular data (Vero et al., 2023) was recently severely violated by the introduction of gradient inversion attacks in the honest-but-curious setting. In these attacks, the FL server infers the client data based on passively observed client gradients and the models where they were computed. However, no prior work investigated the vulnerability of GNNs to such attacks.

**This work: Gradient inversion attack on graphs**   In this work, we introduce the first gradient inversion attack on graphs called *Graph Reconstruction Algorithm for Inversion of Gradients* (**GRAIN**), specifically designed to attack GNNs by recovering both the graph structure and the node features. At the core of GRAIN is an efficient filtering mechanism to correctly identify likely subgraphs, which are then combined to reconstruct the entire graph. In particular, we leverage span checks to exploit the rank-deficiency of GNN layer updates and recover both the discrete set of per-layer node features and the subgraph adjacency matrices. We then reconstruct the client input using a depth-first search (DFS) algorithm to piece together the full graph from the recovered subgraphs.

We evaluate our attack on real-world chemical, citation and social network datasets, achieving reconstruction accuracies of up to 75% (exact) and 85% (partial) on chemical graph classification, 61% in citation graph classification, and 66% in molecular node classification tasks with known node labels. Finally, we demonstrate that data-dependent traversal strategies allow GRAIN to scale to significantly larger graphs, recovering 85% of graphs with around 25 nodes.

**Main Contributions**    Our main contributions are:

- The first gradient inversion attack on Graph Neural Networks, recovering both the graph structure and the node features. We provide an efficient implementation on GitHub.[1]
- A generalization of the theory presented by Petrov et al. (2024) facilitating an effective mechanism to recover individual feature vectors and thus enabling the use on GNN layers to recover the graph connectivity via efficient filtering.
- A novel set of metrics for measuring the quality of the recovered client graph structure and node features, enabling the evaluation of graph gradient inversion attacks at scale.
- A thorough evaluation of GRAIN showing FL with GNNs does not preserve the client data privacy in realistic applications, as GRAIN often recovers clients' graph data exactly.

We believe this work is an important step to further quantify the risks of using private data in FL.

## 2    RELATED WORK

Gradient inversion attacks (Zhu et al., 2019), are attacks to Federated Learning that aim to infer the client's private data from the FL updates clients share with the federated server. As such, they assume knowledge of the updates themselves, as well as the model weights on which the updates were computed. Depending on the attack model, gradient inversion attacks are either malicious (Boenisch et al., 2021; Fowl et al., 2022b;a; Chu et al., 2023; Wen et al., 2022) if the attacker can additionally manipulate the model weights sent to the clients, or honest-but-curious (Zhu et al., 2019; Phong et al., 2018; Zhao et al., 2020; Geiping et al., 2020; Geng et al., 2021; Zhang et al., 2023; Li et al., 2022; Deng et al., 2021; Balunovic et al., 2022; Dimitrov et al., 2024; Petrov et al., 2024; Vero et al., 2023) if the attack is executed passively by just observing model weights and updates.

In this work, we focus on the harder setting of honest-but-curious gradient inversion attacks. Most existing honest-but-curious attacks formulate gradient inversion as an optimization problem (Zhao et al., 2020; Geiping et al., 2020; Yin et al., 2021; Geng et al., 2021; Zhang et al., 2023; Li et al., 2022; Deng et al., 2021; Balunovic et al., 2022) where the attacker tries to obtain the data which corresponds to a client update that matches the observed one best. While this approach is effective in many domains like images (Geiping et al., 2020; Yin et al., 2021; Geng et al., 2021; Zhang et al., 2023; Li et al., 2022) where the client data is continuous, it has been shown that the associated optimization problem is much harder to solve for domains where client inputs are discrete. Some prior works have attempted to alleviate this issue by relying on various continuous relaxation (Balunovic et al., 2022; Vero et al., 2023) to the discrete optimization problem with some success.

In contrast to such approaches, recent research has demonstrated that exact gradient inversion is possible for both continuous (Dimitrov et al., 2024) and discrete inputs (Petrov et al., 2024) in certain neural architectures. Notably, DAGER (Petrov et al., 2024) showed that when dealing with a large but countable number of options for the client input data, the low-rank structure of gradient updates in fully connected layers can be leveraged to efficiently test all possibilities and identify the true input data. GRAIN extends this theory to GNN layers, exploiting the discrete nature of the unknown to the attacker adjacency matrix $A$ to simultaneously recover the client input features and graph structure, under the assumption of discrete input features. This addresses a critical challenge in graph-specific gradient inversion, where the interdependence between the recovery of the client feature matrix $X$ and the adjacency matrix $A$ renders traditional optimization-based attacks ineffective. Unlike DAGER, however, the structure of GNNs only allows for the recovery of the local graph structure using this approach. To overcome this, we further introduce a DFS-based algorithm that combines local graph structures into a single graph, enabling the recovery of the full client input data.

## 3    BACKGROUND AND NOTATION

**Threat Model**    GRAIN is a honest-but-curious gradient inversion attack executed by a malicious FL server that aims to recover the clients' private data. As such, the server is assumed to know the weight updates sent to clients and the corresponding responses received from them. Following most existing gradient inversion attacks (Deng et al., 2021; Balunovic et al., 2022; Vero et al., 2023;

---

[1]https://github.com/insait-institute/GRAIN

Geiping et al., 2020), we also assume knowledge of the client data structure, including the semantic meaning, value ranges, and normalization of individual input features. This is well-justified as the server needs to enforce consistency in input representations across clients to ensure correct training.

As GRAIN represents the first gradient inversion attack on GNNs, it targets the most traditional federated protocol, FedSGD (McMahan et al., 2017), and the most commonly used GNN architectures — GCN and GAT. Further, as GRAIN is based on our extension to Thm. 3.1, introduced by Petrov et al. (2024), it makes two additional assumptions: (i) the number of nodes in the client graphs is smaller than the embedding dimension of the client GNN layers; and (ii) all input node features in the client graphs are discrete. We denote by $m$ the total number of features, by $\mathcal{F}_i$ the set of possible values for the $i$-th feature, and by $\mathcal{F} = \mathcal{F}_1 \times \mathcal{F}_2 \times \cdots \times \mathcal{F}_m$ the set of all possible feature vectors. As we show in Sec. 6, these assumptions cover many realistic use cases of GNNs.

**Graph Terminology**   Next, we introduce our graph notations. By $\mathcal{V}$ we denote the set of possible graph nodes, where each node $v \in \mathcal{V}$ is associated with a given feature vector. For an undirected graph $\mathcal{G} = (V, E)$ with node set $V \subset \mathcal{V}$ of size $n = |V|$ and edge set $E$, we denote the degree of a node $v \in V$ with $\deg_{\mathcal{G}}(v)$. Further, for a pair of vertices $v_s, v_e \in V$, the distance $\mathrm{dist}(v_s, v_e)$ denotes the number of edges in the shortest path connecting $v_s$ to $v_e$. We introduce the notion of a $k$-hop neighborhood of a node $v$, defined by the subgraph $\mathcal{N}_{\mathcal{G}}^k(v) = (V_v^k, E_v^k) \subset \mathcal{G}$ consisting of all nodes $V_v^k = \{v' \in V \mid \mathrm{dist}(v, v') \leq k\}$ in the graph at a distance $\leq k$ from $v$ and the edges between them that can be traversed from $v$ in $\leq k$ steps $E_v^k = \{e = (v_1, v_2) \in E \mid v_1 \in V(\mathcal{N}_{\mathcal{G}}^{k-1}(v)), v_2 \in V_v^k\}$ with $\mathcal{N}_{\mathcal{G}}^0(v) = \{v\}$. Finally, we will call a triplet $(V_v^k, E_v^k, v)$ associated with the $k$-hop neighborhood around $v$ in $\mathcal{G}$ *the building block $\mathcal{G}_v^k$ with center $v$.*

**Graph Neural Networks**   Graph Neural Networks (GNNs) extend traditional neural networks to handle graph-structured data by leveraging the edges between nodes through message passing. Each GNN layer captures complex relationships between nodes by combining information from their neighbors and the graph's structural properties. This allows the model to learn richer node embeddings and gain insights into the graph's topology. In particular, the $l^{\text{th}}$ GNN layer takes as an input a matrix $\boldsymbol{X}^l \in \mathbb{R}^{n \times d}$ of $d$-dimensional node features for each node $v \in V$ and performs a combination of messages passing and non-linearity to produce the node features of the next layer $\boldsymbol{X}^{l+1}$:

$$\boldsymbol{X}^{l+1} = \sigma(\boldsymbol{Z}^l) = \sigma(\boldsymbol{A}^l \boldsymbol{Y}^l) = \sigma\left(\boldsymbol{A}^l \boldsymbol{X}^l \boldsymbol{W}^l\right), \tag{1}$$

where $\boldsymbol{A}^l \in \mathbb{R}^{n \times n}$ is a weighted adjacency matrix, $\boldsymbol{W}^l \in \mathbb{R}^{d \times d'}$ is the weight matrix, $\boldsymbol{Y}^l \in \mathbb{R}^{n \times d'}$ is the output to the linear layer, and $\sigma$ is an activation function. For GCNs, the adjacency weights are calculated using the respective node degrees, while for GATs they are determined by the attention mechanism. We denote the input features matrix with $\boldsymbol{X}^0$, and abuse the notation $\boldsymbol{X}_v^l$ to denote the row of $\boldsymbol{X}^l$ corresponding to the node $v \in V$. We consider $L$-layer GNNs, where we denote with $f_l$ the function that maps the input graph to the output of the $l^{\text{th}}$ layer for $l = 0, 1, 2, \ldots, L-1$.

**Gradient Filtering in Linear Layers**   Recently, DAGER (Petrov et al., 2024) showed that one can leverage the gradients of the network loss $\mathcal{L}$ w.r.t. the weights $\boldsymbol{W}^l$ of the $l^{\text{th}}$ linear layer $\frac{\partial \mathcal{L}}{\partial \boldsymbol{W}^l}$ to search for the correct set of inputs $\boldsymbol{X}^l$ to the layer among a discrete set of possibilities via filtering enabled by the low-rankness of the weight updates. We restate the theoretical findings below:

**Theorem 3.1.** *If $n < d$ and if the matrix $\frac{\partial \mathcal{L}}{\partial \boldsymbol{Y}^l}$ is of full rank, then $\mathrm{rowspan}(\boldsymbol{X}^l) = \mathrm{colspan}(\frac{\partial \mathcal{L}}{\partial \boldsymbol{W}^l})$.*

To verify whether an input vector $\boldsymbol{z}$ can be a part of the client input, DAGER performs a spancheck by measuring the distance between $\boldsymbol{z}$ and the subspace spanned by the column vectors of $\frac{\partial \mathcal{L}}{\partial \boldsymbol{W}^l}$:

$$d(\boldsymbol{z}, \tfrac{\partial \mathcal{L}}{\partial \boldsymbol{W}^l}) := \|\boldsymbol{z} - \mathrm{proj}(\boldsymbol{z}, \mathrm{colspan}(\tfrac{\partial \mathcal{L}}{\partial \boldsymbol{W}^l}))\|_2.$$

We say $\boldsymbol{z}$ can be a part of the $l$-th layer input if $d(\boldsymbol{z}, \frac{\partial \mathcal{L}}{\partial \boldsymbol{W}^l}) < \tau$ for a chosen threshold $\tau$. In our work, we will extend Thm. 3.1 to Eq. 1, in particular applying it to the linear layer $\boldsymbol{Y}^l = \boldsymbol{X}^l \boldsymbol{W}^l$.

## 4   OVERVIEW OF GRAIN

Next, we provide a high-level overview of GRAIN, visualized in Fig. 1. GRAIN a gradient inversion attack designed to reconstruct graph-structured client training data in FL assuming an honest-but-curious adversary and is based on the key observation that due to the architecture of GNNs, the

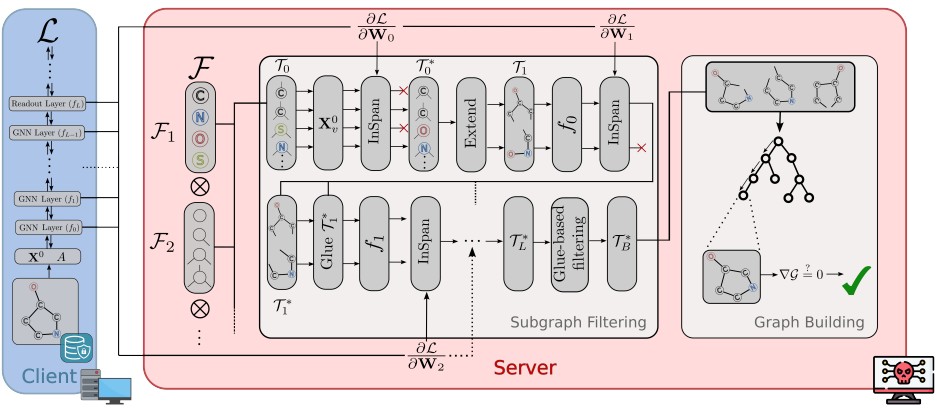

Figure 1: Overview of GRAIN. GRAIN first recovers the input nodes $\mathcal{T}_0^*$ by filtering through the cross-product $\mathcal{T}_0$ of all possible feature values, e.g., all atom types $\mathcal{F}_1$ and all number of bonds $\mathcal{F}_2$. It then iteratively combines and filters them into a set of larger building blocks $\mathcal{T}_B^*$ up to a degree $L$. Finally, it reconstructs the input graph by combining building blocks from $\mathcal{T}_B^*$ in a DFS manner.

input embedding of a node $v$ at layer $l$ can be influenced only by the original input embeddings of the nodes in the $l$-hop neighborhood of $v$. Therefore, given any $l$-hop neighborhood, we can obtain the corresponding embedding of its center $v$ and use our spancheck, inspired by Thm. 3.1 and shown in Thm. 5.1, to check whether the neighborhood is a possible subgraph of the input graph $\mathcal{G}$ or not.

**Filtering** We leverage this subgraph checking procedure to create the filtering stage of GRAIN. In particular, we first generate the node proposal set $\mathcal{T}_0$ consisting of all nodes the can be obtained using the known to the attacker sets of possible feature values $\mathcal{F}_i$. We then apply the span checks layer-by-layer starting from $\mathcal{T}_0$. At each layer $l$, we "glue" the $(l-1)$-hop building blocks recovered from the previous filtering iteration into the set of possible $l$-hop neighbourhoods, denoted $\mathcal{T}_l$ in Fig. 1, which are then filtered using Thm. 5.1 to produce the set of consistent $l$-hop building blocks $\mathcal{T}_l^*$. Finally, at layer $L$ a final consistency check is performed to obtain the final set of $L$-hop building blocks $\mathcal{T}_B^*$.

**Graph Building** In the second stage, we perform graph building where we combine the $L$-hop building blocks in $\mathcal{T}_B^*$ using a DFS-based approach to obtain the final graph reconstruction. To do this, at each node of the DFS tree, we "glue" a building block at a graph node that does not yet have enough neighbors to match their degree. Here, we use that the degree of a node is a widely used node feature for training GNNs (Hamilton et al., 2017; Xu et al., 2018; Cui et al., 2022), and is thus known by the attacker at this stage. When we cannot extend the graph further, we compute its gradient and compare it to the client gradient. If they do not match, we backtrack and try a different path. Otherwise, we terminate the DFS successfully and return the reconstructed graph.

## 5 GRAIN: EXACT GRAPH RECONSTRUCTION FROM GRAIDENTS

We now present the technical details of GRAIN. First, in Sec. 5.1, we explain the key operation of graph gluing. Then, in Sec. 5.2, we present Thm. 5.1 that adapts Thm. 3.1 to GNN layers and Thm. 5.3 that enables GRAIN to locally recover graph structures. These theoretical developments allow for the

---

**Algorithm 1** The GRAIN algorithm

1: **function** GRAIN($\mathcal{T}_0, \frac{\partial \mathcal{L}}{\partial \boldsymbol{W}}, \tau, \boldsymbol{f}, \mathcal{C}$)
2:     $\mathcal{T}_L^* \leftarrow$ GENERATEBBS($\mathcal{T}_0, \frac{\partial \mathcal{L}}{\partial \boldsymbol{W}}, \tau, \boldsymbol{f}$)
3:     $\mathcal{T}_B^* \leftarrow$ STRUCTUREFILTER($\mathcal{T}_L^*, \frac{\partial \mathcal{L}}{\partial \boldsymbol{W}}$)
4:     **return** RECONSTRUCTGRAPH($\mathcal{T}_B^*, \frac{\partial \mathcal{L}}{\partial \boldsymbol{W}}, \mathcal{C}$)

---

efficient removal of proposal elements from $\mathcal{T}_l$, which fail the span check and hence cannot be a subgraph of the input, as we detail in Sec. 5.3. Finally, in Sec. 5.4 we demonstrate our graph building, which recovers the entire graph from the filtered set of possible subgraphs $\mathcal{T}_B^*$ using DFS.

### 5.1 GRAPH GLUING

In this section we describe the process of gluing a $l$-hop building block $\mathcal{B} = (V^B, E^B, c^B)$ to a graph $\mathcal{G} = (V, E)$ at a vertex $c \in V$. The resulting set of graphs $\mathbb{G}$ contains all possible ways of attaching the non-overlapping parts of $\mathcal{B}$ to $\mathcal{G}$ at $c$, as shown in Fig. 2. To this end, we combine the 2 graphs by correctly matching equivalent nodes between them based on their features. In particular,

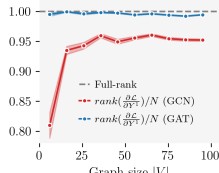
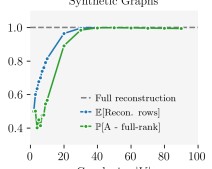
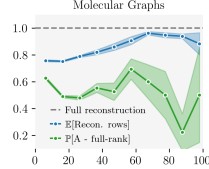
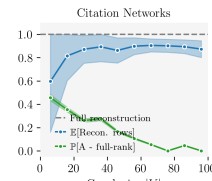

(a) GCN vs GAT reconstructability (on Citeseer)

(b) Impact of low-rankness of the adjacency matrix on reconstructability for GCNs for synthetic data (left), molecules (middle), and citation networks (right)

Figure 3: Ablation studies on how the data and architecture affect reconstructability

we return the empty set if the center of the building block $v'$ and the chosen node $v$ from the graph have different features. The same holds if the $l$-hop neighborhood of the chosen node does not match a subgraph of the building block. In all other cases we return the set of all possible graphs resulting from the gluing $\mathbb{G} = \text{glue}(\mathcal{G}, \mathcal{B}, v)$. We describe how to efficiently perform gluing in App. A.3.

## 5.2 THEORETICAL FOUNDATIONS OF THE SPANCHECK FILTERING

**Span check for GNN layers** We now state our main result extending Thm. 3.1 to GNN layers. The proof is in App. A.2.

**Theorem 5.1.** *If $n < d$, $\boldsymbol{X}_i^l \in \text{colspan}(\frac{\partial \mathcal{L}}{\partial \boldsymbol{W}^l})$ if and only if $\frac{\partial \mathcal{L}}{\partial \boldsymbol{Y}_i^l} \notin \text{rowspan}(\frac{\partial \hat{\mathcal{L}}}{\partial \boldsymbol{Y}_i^l})$, where $\frac{\partial \hat{\mathcal{L}}}{\partial \boldsymbol{Y}_i^l}$ denotes the matrix $\frac{\partial \mathcal{L}}{\partial \boldsymbol{Y}^l}$ with its $i$-th row removed.*

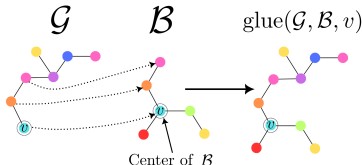

Figure 2: Glueing visualization

This important generalization of the theory presented in DAGER provides an exact condition for recovering individual input vectors $\boldsymbol{X}_i^l$ under any output gradient $\frac{\partial \mathcal{L}}{\partial \boldsymbol{Y}^l}$. In particular, when $\frac{\partial \mathcal{L}}{\partial \boldsymbol{Y}^l}$ is full-rank, the theorem recovers the statement of Thm. 3.1. Otherwise, intuitively, the theorem states that we lose recoverability for inputs $\boldsymbol{X}_i^l$ for which the corresponding row in $\frac{\partial \mathcal{L}}{\partial \boldsymbol{Y}^l}$ is linearly dependent of the rest of the rows in $\frac{\partial \mathcal{L}}{\partial \boldsymbol{Y}^l}$. The emprical experiments, illustrated in Fig. 3a showcase that $\frac{\partial \mathcal{L}}{\partial \boldsymbol{Y}^l}$ is almost certainly full-rank for GATs, enabling the span check to accurately filter the entire input, making Thm. 3.1 still applicable most of the time. However, in general $\frac{\partial \mathcal{L}}{\partial \boldsymbol{Y}^l}$ can exhibit low-rankness, which limits the recovery of the entire input. In particular, Fig. 3b shows that for GCNs for small graphs the full-rankness assumption is violated. To this end, we present the following corollary (proven in App. A.2) which outlines the implications of Thm. 5.1 specifically for GCNs:

**Corollary 5.2.** *For $\frac{\partial \mathcal{L}}{\partial \boldsymbol{Z}^l}$ of full-rank, $n < d$, if the (possibly normalized) adjacency matrix at layer $l$, $\boldsymbol{A} \in \mathbb{R}^{n \times n}$, $\boldsymbol{X}_i^l \in \text{colspan}(\frac{\partial \mathcal{L}}{\partial \boldsymbol{W}^l})$ if and only if $\boldsymbol{A}_i^T \notin \text{colspan}(\hat{\boldsymbol{A}}_i)$. Further, if $\boldsymbol{A}$ is full-rank, then $\boldsymbol{X}_i^l \in \text{colspan}(\frac{\partial \mathcal{L}}{\partial \boldsymbol{W}^l})$ for all $i = 1, 2, \ldots, n$.*

i.e. when $\boldsymbol{A}$ is full-rank, all feature vectors at layer $l$ are recoverable via the spancheck. If this isn't the case, according to the main theorem we still recover most, but not all. In Fig. 3b and App. C.3 we demonstrate that even when $\boldsymbol{A}$ is significantly rank-deficient, the majority of $\boldsymbol{X}^l$ can still be recovered under the GCN setting. Our experiments on real-world datasets show that even in these cases GRAIN is able to partially recover the graph. Next we explain how we propagate our proposed building blocks through the GNN layers in order to be able to apply Thm. 5.1 to filter them.

**Building block propagation** In order to be able to apply Thm. 5.1 to filter a generated $l$-hop building block at the $l$-th layer we need to reconstruct its corresponding input embedding vector $\boldsymbol{X}_j^l$ to the $l$-th layer. We now state our key result allowing us to recover this input vector.

**Theorem 5.3.** *For GNNs satisfying Asm. A.1, propagating a correctly reconstructed building block $\mathcal{G}_v^l$ centred at $v$ through the first $l$ GNN layers recovers the original embedding of $v$ at layer $l$:*

$$f_{l-1}(\mathcal{G}_v^l)[j] = X_i^l,$$

*where $i$ is the index of $v$ in the adjacency matrix of the original input graph $\mathcal{G}$ and $j$ is the index of $v$ in the adjacency matrix of $\mathcal{G}_v^l$.*

Intuitively, this shows that a correctly recovered $l$-hop neighborhood is sufficient to compute the embeddings of $v$ at layer $l$, enabling the span check filter at this layer. The proof is available in App. A.2, alongside a proof that GCNs and GATs satisfy Asm. A.1.

### 5.3 Subgraph filtering

Next, we begin our attack with the creation of the $L$-hop building blocks and reduce the search space via the span check mechanism. Even though GRAIN is applicable to any type of discrete features, we describe our methodology for one-hot encoded ones for notational convenience.

**Recovering the single node feature vectors in $\mathcal{T}_0^*$** We now describe how to recover the single node client feature vectors, that is the 0-hop neighborhoods in $\mathcal{T}_0^*$. We note that the direct enumeration approach by DAGER is often infeasible for graph data. To this end, instead of performing span checks on the entire input vectors, we do partial span checks on partial vectors, iteratively adding dimensions in order to control the complexity. Specifically, we first take the dimensions corresponding to a subset of features and filter those. To further constrain the search space, we then keep adding dimensions corresponding other features, followed by another filtering. We note the soundness of this partial filtering is covered by Thm. 5.1, as performing the spancheck on a subspace is less restrictive compared to the full span check. Pseudocode and further explanations are provided App. A.4.

**Creating 1-hop building blocks $\mathcal{T}_1^*$** We first define the extension $\text{ext}(v)$ of a node $v$ to be the set of all 1-hop building blocks that can be constructed by attaching exactly $\deg(v)$ nodes from $\mathcal{T}_0^*$ to $v$. We note that we do not attach nodes $w \in \mathcal{T}_0^*$ to $v$ if the feature degree of $w$ is 0, that is $\deg(w) = 0$. The set of all possible 1-hop building blocks $\mathcal{T}_1 = \bigcup_{v \in \mathcal{T}_0^*} \text{ext}(v)$ is then defined as the set of all possible 1-hop neighborhoods that can be constructed by extending node from $\mathcal{T}_0^*$. Thm. 5.3 then allows us to exactly recover the first

---

**Algorithm 2** Filtering using the spancheck

1: **function** FILTER($\mathcal{T}_l, \frac{\partial \mathcal{L}}{\partial \boldsymbol{W}^l}, \tau, f_{l-1}$)
2:      $\mathcal{T}_l^* \leftarrow \{\}$
3:      **for** $\mathcal{G}$ in $\mathcal{T}_l$ **do**
4:          $v \leftarrow \text{center}(\mathcal{G})$
5:          **if** $d(f_{l-1}(\mathcal{G})_v, \frac{\partial \mathcal{L}}{\partial \boldsymbol{W}^l}) < \tau$ **then**
6:              $\mathcal{T}_l^* \leftarrow \mathcal{T}_l^* \cup \{\mathcal{G}\}$
7:      **return** $\mathcal{T}_L^*$

---

layer embedding for the center of each neighborhood. We are hence able to filter $\mathcal{T}_1$ by applying Thm. 5.1 on $\frac{\partial \mathcal{L}}{\partial \boldsymbol{W}^1}$ to achieve the reduced set of 1-hop building blocks $\mathcal{T}_1^*$, as shown in Alg. 2.

**Creating $l$-hop building blocks $\mathcal{T}_l^*$** For a building block $\mathcal{B} \in \mathcal{T}_l^*$, we define the *dangling nodes* $\text{dang}(\mathcal{B})$ as the set of all nodes $v \in \mathcal{B}$ such that $\deg(v)$ (the ground-truth degree of the node) is greater than the number of its neighbors. We extend the $l$-hop building blocks $\mathcal{B} \in \mathcal{T}_l^*$ by calculating all possible gluings of 1-hop building blocks $\mathcal{B}' \in \mathcal{T}_1^*$ to all dangling nodes of $\mathcal{B}$. This is shown in lines 10–16 of Alg. 3. The resulting set is then called $\mathcal{T}_{l+1}$, which is then filtered by applying Thm. 5.1 on $\frac{\partial \mathcal{L}}{\partial \boldsymbol{W}^{l+1}}$ to achieve the reduced set of $l+1$-hop building blocks $\mathcal{T}_{l+1}^*$. We repeat the process explained above until we reach the desired $L$-hop neighborhoods, with the final spancheck performed on the first linear layer of the commonly used readout classification head.

---

**Algorithm 3** Creating the degree-$L$ building blocks

1: **function** GENERATEBBS($\{\mathcal{F}_i\}_{i \in \{1,f\}}, \frac{\partial \mathcal{L}}{\partial \boldsymbol{W}}, \tau, \boldsymbol{f}$)
2:      $\mathcal{T}_0^* \leftarrow \text{FILTERNODES}(\{\mathcal{F}_i\}, \frac{\partial \mathcal{L}}{\partial \boldsymbol{W}_0}, \tau)$
3:      $\mathcal{T}_1 \leftarrow \{\}$
4:      **for** $v$ in $\mathcal{T}_0^*$ **do**
5:          $\mathcal{T}_1 \leftarrow \mathcal{T}_1 \cup \text{ext}(v, \mathcal{T}_0^*)$
6:      $\mathcal{T}_1^* \leftarrow \text{FILTER}(\mathcal{T}_1, \frac{\partial \mathcal{L}}{\partial \boldsymbol{W}_1}, \tau, f_0)$
7:      **for** $l \leftarrow 1, \ldots, L-1$ **do**
8:          $\mathcal{T}_{l+1} \leftarrow \{\}$
9:          **for** $G$ in $\mathcal{T}_l^*$ **do**
10:             $S \leftarrow \{G\}$
11:             **for** $v$ in $\text{dang}(G)$ **do**
12:                 $S' \leftarrow \{\}$
13:                 **for** $\mathcal{G}', \mathcal{G}^B$ in $S \times \mathcal{T}_1^*$ **do**
14:                     $S' \leftarrow S' \cup \text{GLUE}(\mathcal{G}, \mathcal{G}^B, v, l)$
15:                 $S \leftarrow S'$
16:             $\mathcal{T}_{l+1} \leftarrow \mathcal{T}_{l+1} \cup S$
17:          $\mathcal{T}_{l+1}^* \leftarrow \text{FILTER}(\mathcal{T}_{l+1}, \frac{\partial \mathcal{L}}{\partial \boldsymbol{W}_{l+1}}, \tau, f_{l-1})$
18:      **return** $\mathcal{T}_L^*$

---

**Additional structure-based filtering** To further restrict the proposal set of building blocks, we perform a consistency check to rule out blocks that cannot be part of the ground truth graph. Specifically, for every building block $\mathcal{B} \in \mathcal{T}_L^*$ and for every dangling node in it $v \in \text{dang}(\mathcal{B})$ we assert that there exists a building block in $\mathcal{T}_L^*$ that we can glue it at $v$ to $\mathcal{B}$. If this is not the case, we know that either $\mathcal{B}$ is the input graph (which we check by computing the gradients produced by $\mathcal{B}$), or it cannot be part of the ground truth graph and remove it from $\mathcal{T}_L^*$. We denote the resulting set $\mathcal{T}_B^*$.

## 5.4 GRAPH BUILDING

Finally, we describe how we leverage depth-first search to combine the filtered set of building blocks $\mathcal{T}_B^*$ into our final graph reconstruction.

Specifically, each node of our DFS exploration tree represents a partially reconstructed client graph $\mathcal{G}_{\text{curr}}$, and at each branch we choose an arbitrary dangling node $v \in \mathcal{G}_{\text{curr}}$ (Line 7 in Alg. 4) and generate all possible graphs $\mathbb{G} = \text{glue}(\mathcal{G}_{\text{curr}}, \mathcal{B}, v)$ that can be created by gluing a building block $\mathcal{B} \in \mathcal{T}_B^*$ at $v$ (Line 8 in Alg. 4). This is done by iterating over all the building blocks in $\mathcal{T}_B^*$, checking for each one if it is possible to glue it to $\mathcal{G}_{\text{curr}}$ and if so, saving the extended graph as a new branch. The pseudocode for the branching is provided in App. A.6. During branching, we also take care of recovering possible cycles within the recon-

**Algorithm 4** DFS reconstruction

1: **function** DoDFS($\mathcal{T}_B^*, \frac{\partial \mathcal{L}}{\partial \boldsymbol{W}}, \mathcal{G}_{\text{curr}}, \mathcal{C}$)
2:     **if** $|dang(\mathcal{G}_{\text{curr}})| == 0$ **then**
3:         **return** $\Delta_{\mathcal{G}_{\text{curr}}}, \mathcal{G}_{\text{curr}}$
4:
5:     $\mathcal{G}_{\text{TOP}}, d_{\text{TOP}} \leftarrow \emptyset, \infty$
6:     $\mathbb{B}_{\text{ord}} \leftarrow \text{Order}(\mathcal{T}_B^*)$
7:     $v \leftarrow Sample(\{v \in dang(G_{\text{curr}})\})$
8:     $\mathbb{G}_{\text{new}} \leftarrow \text{Branch}(\mathbb{B}_{\text{ord}}, \mathcal{G}_{\text{curr}}, v)$
9:     **for** $\mathcal{G}$ in $\mathbb{G}_{\text{new}}$ **do**
10:         $d', \mathcal{G}' \leftarrow \text{DoDFS}(\mathcal{T}_B^*, \frac{\partial \mathcal{L}}{\partial \boldsymbol{W}}, \mathcal{G}, \mathcal{C})$
11:         **if** $d' == 0$ **then**
12:             **return** $0, \mathcal{G}'$
13:         **else if** $d_{\text{TOP}} > d'$ **then**
14:             $d_{\text{TOP}}, \mathcal{G}_{\text{TOP}} \leftarrow d', \mathcal{G}'$
15:     **return** $d_{\text{TOP}}, \mathcal{G}_{\text{TOP}}$

structions by overlapping nodes in $\mathcal{G}_{\text{curr}}$ that might coincide. This is also elaborated on in App. A.6. If $\mathcal{G}_{\text{curr}}$ has no more dangling nodes, we calculate the distance between its and the client gradients:

$$\Delta_{\mathcal{G}} = \min_{c \in \mathcal{C}} \left\| \frac{\partial \mathcal{L}(\mathcal{G}, c)}{\partial \boldsymbol{W}} - \frac{\partial \mathcal{L}}{\partial \boldsymbol{W}} \right\|_F, \tag{2}$$

where $\mathcal{C}$ is the set of all possible labels and $\| \cdot \|_F$ is the standard Frobenius norm. If $\Delta_{\mathcal{G}_{\text{curr}}} = 0$ the algorithm terminates early with correct graph (Line 12 in Alg. 4). Otherwise, we return the best possible reconstruction in gradient distance (Lines 9-14 in Alg. 4).

**Building block ordering** A key factor in the performance of the algorithm is the order in which we visit nodes of the exploration tree. We use heuristic ordering based on a score $S(\mathcal{B})$ we define for every building block $\mathcal{B} \in \mathcal{T}_B^*$. We first define the score $S_v(\mathcal{B})$ to be equal to the lowest span check distance $d(\mathcal{G}, \frac{\partial \mathcal{L}}{\partial \boldsymbol{W}^L})$ of a building block $\mathcal{B}$ that can be glued to $\mathcal{G}$ at $v$. The score for the entire block is then calculated as the sum of the vertex scores $S(\mathcal{B}) = \sum_v S_v(\mathcal{B})$. Intuitively, this score represents how compatible any building block from $\mathcal{T}_B^*$ is with the other building blocks that passed the spanchecks. Again, intuitively, the building blocks which are part of the input will be more compatible with the other building blocks from the input than potential false positives.

**Uniqueness heuristic** In some settings, such as social or citation networks we notice that the feature vectors of different nodes are almost surely unique. In these settings, we leverage a heuristic which during reconstruction always overlaps any two nodes with the same features. Further, in these settings we never try to glue the same building block twice, as $L$-hop neighborhoods are analogically unique. The heuristic drastically reduces the search space and the time for convergence of GRAIN.

## 6 EVALUATION

In this section we evaluate GRAIN's performance against prior gradient leakage methods.

We begin by detailing our experimental setup and the baseline attacks considered, along with introducing a novel set of metrics, specifically designed to jointly assess the differences in node features and graph topology between the true client graphs and their reconstructions. The experimental results demonstrate GRAIN's substantial improvements over existing attacks, achieving superior reconstruction accuracy and versatility. Namely, we demonstrate that GRAIN remains effective across a wide range of architectural changes, GNN model types, data modalities, and task scenarios.

### 6.1 EVALUATION METRIC

We found it necessary to design our own set of metrics, as prior graph-related similarity measurements were not suitable for evaluating gradient inversion attacks. A discussion on the desired qualities of the metrics, and the reasoning behind their design can be found in App. B.

To this end, we introduce the ***Graph Similarity Metrics*** (GSM) - a set of metrics designed to evaluate the similarity of a pair of graphs $\mathcal{G}$ and $\hat{\mathcal{G}}$ under the name GSM-N, the details of which we showcase

in App. B. A key highlight of GSM is that it assesses structures of varying globality, while ensuring the metric is invariant to graph isomorphism through rigorous node matching.

We utilise 3 separate instances of the metric - namely for $N = 0, 1, 2$, where larger-hop neighborhoods are used to capture more structural information. It is important to note that all measurements are scaled by a factor of $\frac{\min(|\mathcal{V}|, |\hat{\mathcal{V}}|)}{\max(|\mathcal{V}|, |\hat{\mathcal{V}}|)}$ to penalize reconstructions of incorrect size. We further report the percentage of exactly reconstructed graphs, denoted by FULL in the result tables.

To measure the perceived reconstruction quality and compare it to our GSM set of metrics, as well as to confirm that our metrics are fair with respect to the baseline attacks, we further conducted a human evaluation study. In Tab. 1, we show that our metrics are highly correlated with human perception. Further details about how this study was conducted are shown in App. C.

## 6.2 Experimental setup

Next, we describe our experimental setup, including the architecture of the attacked models, the client datasets used, and the hardware required by the attacker.

Table 1: Comparison of GSM and human evaluation.

|       | GSM-0 | GSM-1 | GSM-2 | Human |
|-------|-------|-------|-------|-------|
| GRAIN | **72.6** | **67.8** | **66.9** | **70.6** |
| DLG   | 24.2  | 10.5  | 12.0  | 6.5   |

**Architecture details**  Unless otherwise specified, all of our attacks are applied on 2-layer GNNs ($L = 2$) with a hidden embedding dimension $d' = 300$ and a ReLU activation. For GAT experiments 2-headed attention was used (adapting GRAIN to GATs with more heads is analogous). All networks also feature a 2-layer feedforward network for performing the readout — a common depth for GNNs Kipf & Welling (2016). Given the depth restrictions, we recover building blocks up to layer 2, with the first readout layer being used for the relevant filtering of the largest blocks. In Tab. 10, we show that our attack is robust with respect to changes in these architectural parameters.

**Evaluation datasets**  We evaluate on three different types of graph data – chemical data, citation and social networks. For the chemical experiments, we evaluate on molecule property prediction data, where molecules are represented as graphs and each node is a given atom. We follow the common convention to omit hydrogen atoms in the graphs. Each node is embedded by concatenating the one-hot encodings of 8 features (Xu et al., 2018; Wu et al., 2020), namely the atom type, formal charge, number of bonds, chirality, number of bonded hydrogen atoms, atomic mass, aromaticity and hybridization (Rong et al., 2020). We evaluate GRAIN on 3 well-known chemical datasets – Tox21, Clintox, and BBBP, introduced by the **MoleculeNet** benchmark (Wu et al., 2018).

For the citation networks experiments, we apply GRAIN on the CiteSeer(Giles et al., 1998) dataset, which features a single graph with 3312 nodes representing scientific publications classified into one of six classes. The edges between them represent citations, and the features of any node is a 0/1-valued word vector of length 3703 indicating the absence/presence of a keyword in the abstract.

Finally, for the social network experiments we use the Pokec(Rossi & Ahmed, 2015) dataset, containing 1.6 million nodes (users), where connections represent friendship between users. We chose discrete node features where every feature with frequency less that $1\%$ was categorized as "Other", as these entries usually contain irrelevant outliers. This resulted in 36 total discrete features.

To simulate a federated learning environment, in the latter two settings we sample subgraphs of a given size from the datasets for each of the FL clients and use cluster classification objective, where each subgraph is classified as the most common class among the comprising nodes. We chose the sampling distribution: 20 graphs with 1-10 nodes, 40 graphs with 10-20 nodes, 30 graphs with 20-30 nodes, and 10 graphs with 30-40 nodes, closely mirroring the distribution of the Tox21 dataset.

**Computational requirements**  We provide an efficient GPU imlementation, where each experiment has been run on a NVIDIA L4 Tensor Core GPU with less than 40GB of CPU memory.

## 6.3 Baseline Attacks

We adapt the DLG attack (Zhu et al., 2019), a standard continuous attack, and TabLeak, an attack purposefully designed for recovering discrete tabular data. As described in (Vero et al., 2023), all input features are first passed through an initial sigmoid layer to ensure they are in the interval (0,

Table 2: Results (in %) of experiments on the 3 dataset types – Tox21 (chemical), CiteSeer (citation), Pokec (social). Here "$+A$" refers to the baseline attack with the input adjacency matrix given.

| | | GCN | | | | | GAT | | | | |
|---|---|---|---|---|---|---|---|---|---|---|---|
| | | GSM-0 | GSM-1 | GSM-2 | FULL | Time [h] | GSM-0 | GSM-1 | GSM-2 | FULL | Time [h] |
| Tox21 | GRAIN | $86.9^{+4.2}_{-5.7}$ | $83.9^{+5.2}_{-6.9}$ | $82.6^{+5.7}_{-7.4}$ | $68.0 \pm 1.7$ | 14.3 | $92.9^{+3.8}_{-5.8}$ | $90.7^{+5.0}_{-7.1}$ | $89.9^{+5.8}_{-7.2}$ | $75.0 \pm 1.8$ | 10.8 |
| | DLG | $31.8^{+4.5}_{-4.3}$ | $20.3^{+5.5}_{-4.8}$ | $22.8^{+6.6}_{-5.6}$ | $1.0 \pm 0.2$ | 3.3 | $96.0 \pm 0.32$ | $9.3^{+4.4}_{-4.9}$ | $6.5^{+3.9}_{-4.1}$ | $2.0 \pm 0.3$ | **4.2** |
| | DLG $+A$ | $54.7^{+3.9}_{-4.2}$ | $60.1^{+4.6}_{-5.2}$ | $76.7^{+3.6}_{-4.8}$ | $1.0 \pm 0.2$ | **3.1** | $96.5 \pm 0.34$ | $69.7^{+4.1}_{-4.2}$ | $81.3^{+3.4}_{-3.6}$ | $2.0 \pm 0.3$ | 4.5 |
| | TabLeak | $25.1^{+5.1}_{-4.3}$ | $12.4^{+5.5}_{-4.3}$ | $10.8^{+5.6}_{-3.9}$ | $1.0 \pm 0.2$ | 13.1 | $73.7^{+2.6}_{-2.0}$ | $7.2^{+5.2}_{-4.9}$ | $10.0 \pm 4.8$ | $1.0 \pm 0.2$ | 6.0 |
| | TabLeak $+A$ | $55.6^{+3.9}_{-3.9}$ | $57.7^{+4.1}_{-4.6}$ | $73.8^{+2.8}_{-3.5}$ | $1.0 \pm 0.2$ | 12.3 | $75.1^{+2.5}_{-1.9}$ | $74.9^{+2.1}_{-1.9}$ | $84.2^{+1.5}_{-1.3}$ | $1.0 \pm 0.2$ | 6.0 |
| CiteSeer | GRAIN | $62.5^{+7.7}_{-8.2}$ | $31.0^{+8.0}_{-7.8}$ | $31.6^{+8.1}_{-8.1}$ | $20.0 \pm 0.8$ | 2.5 | $79.3^{+4.7}_{-6.3}$ | $69.1^{+6.1}_{-6.4}$ | $69.6^{+6.2}_{-6.0}$ | $61.0 \pm 1.6$ | 2.1 |
| | DLG | $65.7^{+1.6}_{-1.7}$ | $0.0^{+0.0}_{-0.0}$ | $0.1^{+0.1}_{-0.1}$ | $0.0 \pm 0.0$ | 25.6 | $65.7^{+1.6}_{-1.6}$ | $0.0^{+0.0}_{-0.0}$ | $0.0^{+0.0}_{-0.0}$ | $0.0 \pm 0.0$ | 26.3 |
| | DLG $+A$ | $65.7^{+1.6}_{-1.6}$ | $0.0^{+0.0}_{-0.0}$ | $0.0^{+0.0}_{-0.0}$ | $0.0 \pm 0.0$ | 26.8 | $65.7^{+1.7}_{-1.6}$ | $0.0^{+0.0}_{-0.0}$ | $0.0^{+0.0}_{-0.0}$ | $0.0 \pm 0.0$ | 28.2 |
| | TabLeak | $65.2^{+2.5}_{-2.4}$ | $0.0^{+0.0}_{-0.0}$ | $0.3^{+0.4}_{-0.3}$ | $0.0 \pm 0.0$ | 172.7 | $65.0^{+2.4}_{-2.3}$ | $0.0^{+0.0}_{-0.0}$ | $0.0^{+0.0}_{-0.0}$ | $0.0 \pm 0.0$ | 177.4 |
| | TabLeak $+A$ | $65.3^{+2.4}_{-2.2}$ | $0.0^{+0.0}_{-0.0}$ | $0.0^{+0.0}_{-0.0}$ | $0.0 \pm 0.0$ | 172.9 | $65.1^{+2.4}_{-2.3}$ | $0.0^{+0.0}_{-0.0}$ | $0.0^{+0.0}_{-0.0}$ | $0.0 \pm 0.0$ | 171.5 |
| Pokec | GRAIN | $58.3^{+5.9}_{-5.9}$ | $30.7^{+7.9}_{-7.8}$ | $35.8^{+8.3}_{-7.9}$ | $15.0 \pm 0.8$ | 0.3 | $97.2^{+1.6}_{-1.9}$ | $93.5^{+3.4}_{-4.2}$ | $96.3^{+1.9}_{-2.3}$ | $79.0 \pm 1.8$ | 0.5 |
| | DLG | $38.1^{+1.2}_{-1.2}$ | $0.0^{+0.0}_{-0.0}$ | $7.9^{+1.8}_{-1.8}$ | $0.0 \pm 0.0$ | 0.6 | $35.8^{+1.2}_{-1.1}$ | $0.0^{+0.0}_{-0.0}$ | $0.1^{+0.1}_{-0.1}$ | $0.0 \pm 0.0$ | 1.2 |
| | DLG $+A$ | $37.4^{+1.2}_{-1.2}$ | $0.2^{+0.3}_{-0.2}$ | $26.3^{+2.1}_{-2.2}$ | $0.0 \pm 0.0$ | 0.5 | $37.7^{+1.1}_{-1.1}$ | $0.1^{+0.1}_{-0.1}$ | $16.9^{+1.9}_{-1.9}$ | $0.0 \pm 0.0$ | 1.0 |
| | TabLeak | $37.1^{+1.4}_{-1.3}$ | $0.1^{+0.1}_{-0.1}$ | $2.9^{+1.5}_{-1.3}$ | $0.0 \pm 0.0$ | 6.6 | $37.0^{+1.1}_{-1.1}$ | $0.0^{+0.0}_{-0.0}$ | $1.1^{+0.9}_{-0.7}$ | $0.0 \pm 0.0$ | 12.3 |
| | TabLeak $+A$ | $37.9^{+1.3}_{-1.3}$ | $0.5^{+0.5}_{-0.4}$ | $20.1^{+2.8}_{-2.9}$ | $0.0 \pm 0.0$ | 5.7 | $37.7^{+1.4}_{-1.4}$ | $0.3^{+0.3}_{-0.2}$ | $23.4^{+1.8}_{-1.8}$ | $0.0 \pm 0.0$ | 11.4 |

| Original | GRAIN | DLG | Tableak |
|---|---|---|---|

Figure 4: Examples molecule reconstructions. Multivalent interactions are not recovered, as they are not considered by the GNN.

1). Similarly, we ensure the adjacency matrix $A$ is symmetric by optimizing over the upper triangle, and apply a softmax operation over the dummy labels to convert them to probabilities. Finally, we generate a prediction graph by connecting all nodes $v_i, v_j$ corresponding to $\sigma(A)_{ij} \geq 0.5$. Additionally, we test both baselines when they are given the correct adjacency matrix $A$. In all cases we provide the attack with the correct number of nodes to ensure that $X$ and $A$ have the correct shape. We demonstrate that, even when the baselines have a significant amount of prior knowledge, GRAIN significantly outperforms them (see Fig. 4 and Tab. 2).

## 6.4 EXPERIMENTAL RESULTS

Next, we evaluate the baselines and GRAIN and show that GRAIN outperforms the existing baselines across all defined metrics. Further, GRAIN is applicable across a variety of datasets and settings, including being depth- and width-agnostic, and far more scalable and robust. In all measurements we quote the mean value of the metric, as well as the 95% confidence interval around it, measured by generating 10,000 random sample sets via bootstrapping.

**Main experiments** We first apply the algorithms DLG, TabLeak and GRAIN to the 3 types of dataset on both the GCN and GAT architectures. We observe in Tab. 2 that GRAIN achieves a much higher partial reconstruction rate (up to 96%) compared to any baseline. This remains true even when the baseline is informed about the input adjacency matrix $A$. Without $A$, baseline performance notably drops with neighborhood size, showing the baselines' inability to recover the structure. Beyond partial reconstruction, GRAIN is further able to recover up to 80% of the dataset exactly, while the baselines achieve this only in the case of very small graphs. Further, GRAIN closely matches TabLeak's runtime and is much faster than all baselines with the node uniqueness heuristic. For visual inspection, we also include a comparison of a fully reconstructed molecule in Fig. 4, with further examples in App. C.

As shown, GRAIN can also be effectively applied to both architectures, consistently achieving a higher score on the GAT architecture. This increase happens because as we showed in Fig. 3a, $\frac{\partial \mathcal{L}}{\partial Y^l}$ is almost certainly full-rank, where Cor. 5.2 enables the direct recovery of all inputs.

For the Tox21 baseline experiments we used a LBFGS optimizer for more stable and higher quality results. However, for CiteSeer and Pokec, where the client input space is much larger, we instead ran SGD due to time constraints. In App. C.3, we show results with LBFGS on 10 times fewer samples.

Table 3: Comparison of GRAIN and baselines on chemical datasets for GCNs on graphs of size $n$.

| | $n \leq 15$ | | | $16 \leq n \leq 25$ | | | $26 \leq n$ | | |
|---|---|---|---|---|---|---|---|---|---|
| | GSM-0 | GSM-2 | FULL | GSM-0 | GSM-2 | FULL | GSM-0 | GSM-2 | FULL |
| GRAIN | $\mathbf{93.0^{+3.4}_{-5.4}}$ | $\mathbf{91.6^{+3.8}_{-6.3}}$ | $\mathbf{81.9 \pm 1.7}$ | $\mathbf{81.7^{+3.9}_{-4.8}}$ | $\mathbf{74.8^{+5.8}_{-6.3}}$ | $\mathbf{43.6 \pm 1.1}$ | $50.1^{+6.8}_{-7.1}$ | $39.2^{+8.5}_{-7.7}$ | $\mathbf{5.1 \pm 0.6}$ |
| DLG | $27.4^{+4.2}_{-3.8}$ | $13.3^{+5.7}_{-4.6}$ | $1.0 \pm 0.2$ | $25.5^{+3.9}_{-3.5}$ | $16.7^{+5.2}_{-4.4}$ | $0.9 \pm 0.2$ | $25.4^{+4.8}_{-4.3}$ | $14.8^{+6.4}_{-5.8}$ | $0.0 \pm 0.0$ |
| DLG $+A$ | $52.1^{+3.1}_{-3.3}$ | $71.3^{+3.1}_{-3.9}$ | $1.0 \pm 0.2$ | $53.7^{+3.2}_{-3.5}$ | $75.3^{+3.0}_{-3.7}$ | $0.9 \pm 0.2$ | $53.0^{+4.3}_{-4.8}$ | $72.6^{+4.1}_{-5.8}$ | $0.0 \pm 0.0$ |
| TabLeak | $30.3^{+5.0}_{-4.4}$ | $15.4^{+5.8}_{-4.8}$ | $1.9 \pm 0.3$ | $15.7^{+3.0}_{-2.2}$ | $2.1^{+2.2}_{-1.1}$ | $0.0 \pm 0.0$ | $13.0^{+3.3}_{-2.3}$ | $2.8^{+4.1}_{-1.9}$ | $0.0 \pm 0.0$ |
| TabLeak $+A$ | $53.9^{+4.0}_{-4.2}$ | $72.9^{+3.4}_{-3.9}$ | $1.9 \pm 0.3$ | $57.1^{+3.1}_{-3.5}$ | $71.4^{+2.9}_{-3.6}$ | $0.0 \pm 0.0$ | $\mathbf{56.1^{+2.9}_{-3.3}}$ | $\mathbf{74.4^{+2.3}_{-3.1}}$ | $0.0 \pm 0.0$ |

Table 4: Reconstruction results in % for GRAIN on Citeseer and GATs for different ablations.

| | GSM-0 | GSM-1 | GSM-2 | FULL |
|---|---|---|---|---|
| Default | $\mathbf{79.3^{+4.7}_{-6.3}}$ | $\mathbf{69.1^{+6.1}_{-6.4}}$ | $\mathbf{69.6^{+6.2}_{-6.0}}$ | $\mathbf{61.0 \pm 1.6}$ |
| No degree | $59.7^{+6.8}_{-7.2}$ | $42.7^{+6.3}_{-6.6}$ | $43.2^{+6.4}_{-6.6}$ | $32.0 \pm 1.1$ |
| No heuristic | $64.6^{+3.5}_{-4.2}$ | $52.1^{+4.7}_{-5.3}$ | $52.4^{+4.6}_{-5.2}$ | $44.0 \pm 1.3$ |

Table 5: Reconstruction results in % for GRAIN on Tox21 for GCNs in various settings

| | GSM-0 | GSM-1 | GSM-2 | FULL |
|---|---|---|---|---|
| Default | $86.9^{+4.2}_{-5.7}$ | $83.9^{+5.2}_{-6.9}$ | $82.6^{+5.7}_{-7.4}$ | $\mathbf{68.0 \pm 1.7}$ |
| $\sigma = $ GELU | $82.0^{+5.3}_{-6.7}$ | $79.1^{+6.0}_{-7.4}$ | $78.4^{+6.2}_{-8.0}$ | $61.0 \pm 1.6$ |
| Pre-trained | $73.5^{+6.4}_{-7.4}$ | $70.0^{+7.3}_{-7.7}$ | $68.6^{+7.6}_{-8.3}$ | $49.0 \pm 1.4$ |
| Node Class. | $\mathbf{88.0^{+3.8}_{-5.4}}$ | $\mathbf{85.5^{+4.6}_{-6.5}}$ | $\mathbf{84.9^{+5.0}_{-6.6}}$ | $66.0 \pm 1.6$ |

**Effect of graph size on reconstruction** In Tab. 3 and Tab. 12 we show how GRAIN performs on graphs of different sizes under the same setting as our main experiments (for GCNs and GATs respectively). In the chemical setting (Tab. 3) molecules are divided into groups of $n \leq 15$, $16 \leq n \leq 25$ or $n \geq 26$ nodes, aggregated across the 3 chemical datasets. We notice that GRAIN significantly outperforms the baselines for smaller graphs, but the performance decreases on the largest groups in both the chemical and the social networks setting due to timeouts (15 minutes) during graph building. That said, the social network setting (Tab. 12), allows us to apply the node uniqueness heuristic, allowing us to scale to much larger graphs of size up to $n = 60$ nodes. Further, our work still manages to reconstruct a fraction of the large graphs exactly, which is impossible for the baseline models, even in the more difficult chemical setting (discussed in App. D).

**Ablation studies** We analyze the impact of our design choices and heuristics on GRAIN in Tab. 4. Removing the node in-degree from the feature set and enumerating all possibilities during filtering still enabled exact recovery of 30% of graphs but caused substantial degradation. A similar, though less severe, drop occurs without the node uniqueness heuristic (Sec. 5.4), highlighting the key role of data-specific heuristics in strengthening the attack. We also show that GRAIN's stability is unaffected by architecture parameters and thresholds. As demonstrated in App. C.3, network width and depth have minimal impact, provided the embedding dimension exceeds the number of graph nodes ($d > n$). Additionally, the $\tau$ threshold remains robust, with values between $10^{-4}$ and $10^{-2}$ yielding nearly identical filtering performance.

**Additional experiments** We provide additional experiments showcasing GRAIN's performance in different miscellaneous settings in Tab. 5. First, we replace the ReLU activation function in the GCN by a GELU and report that GRAIN achieves similar results, showing our flexibility with respect to different activations. Furthermore, while prior work has shown that gradient inversion becomes significantly more difficult on pre-trained models (Geiping et al., 2020), GRAIN still manages to reconstruct around 50% of molecules exactly. Finally, we achieve consistently strong results in the node classification task, assuming ground-truth labels are known, which can be easily recovered using methods like Zhao et al. (2020).

## 7 CONCLUSION

We introduced GRAIN, the first gradient inversion attack for Graph Neural Networks capable of accurately recovering graphs from shared gradients. By leveraging the rank-deficiency of the GNN layers, we developed an efficient framework for extracting and filtering subgraphs of the input graph, which are iteratively combined to reconstruct the original graph.

Our results showed GRAIN achieves an exact reconstruction rate of up to 80% for graph classification. We introduced new metrics to evaluate partial graph reconstructions and demonstrated that GRAIN significantly outperforms prior work. Moreover, GRAIN maintains high reconstruction quality across different architectures, parameters, and settings, and can scale to much larger graphs.

In summary, our paper is the first to demonstrate that GNN training in a federated learning setting poses data privacy risks. We believe that this is a promising initial step towards identifying these vulnerabilities and developing effective defense mechanisms.

ACKNOWLEDGMENTS

INSAIT, Sofia University "St. Kliment Ohridski". Partially funded by the Ministry of Education and Science of Bulgaria's support for INSAIT as part of the Bulgarian National Roadmap for Research Infrastructure.

This project was supported with computational resources provided by Google Cloud Platform (GCP).

This work has been done as part of the EU grant ELSA (European Lighthouse on Secure and Safe AI, grant agreement no. 101070617) . Views and opinions expressed are however those of the authors only and do not necessarily reflect those of the European Union or European Commission. Neither the European Union nor the European Commission can be held responsible for them.

The work has received funding from the Swiss State Secretariat for Education, Research and Innovation (SERI).

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

# A ADDITIONAL TECHNICAL DETAILS

## A.1 TABLE OF NOTATIONS

For convenience, we add a table of notations containing brief definitons for all symbols used in our work.

Table 6: Table of notations used in the technical description of GRAIN.

| Symbol | Definition | Symbol | Definition |
|---|---|---|---|
| $\mathcal{G} = (V, E)$ | Graph with nodes $V$ and edges $E$ | $n$ | # of nodes in the graph |
| $\boldsymbol{A}^l$ | Possibly weighted adjacency matrix at layer $l$ | $\mathrm{dist}(v_s, v_e)$ | # edges in shortest path connecting nodes $v_s, v_e \in V$ |
| $\mathcal{N}^k_{\mathcal{G}}(v)$ | $k$-hop neighborhood in graph $\mathcal{G}$ with center node $v$ | $\mathcal{L}$ | Loss |
| $\deg_{\mathcal{G}}(v)$ | Degree of node $v$ in graph $\mathcal{G}$ | $\deg(v)$ | Degree of node $v$ as given by its feature |
| $\boldsymbol{X}^i$ | Input to the $i$th GNN layer | $\boldsymbol{X}^i_v$ | $i$-th layer input feature of node $v$ |
| $\boldsymbol{W}^i$ | Weights of the $i$-th layer | $d'$ | Hidden dimension size |
| $L$ | Number of GNN layers | $m$ | Number of features |
| $f_i$ | Function mapping the input graph to the output of the $i$-th layer | $\tau$ | Span check distance threshold |
| $\mathcal{F}$ | $\mathcal{F}_1 \times \cdots \times \mathcal{F}_m$ - set of all possible feature combinations | $\mathcal{F}_i$ | Set of values for the $i$-th feature |
| $\mathcal{T}_l$ | Proposal set of $l$-hop building blocks | $\mathcal{T}^*_l$ | Filtered set of $l$-hop building blocks |
| $\mathcal{T}^*_B$ | Final set of filtered building blocks | $\sigma$ | Activation function |
| $\Delta_{\mathcal{G}}$ | Distance between the gradients of $\mathcal{G}$ and observed gradients | $d_{best}$ | Gradient distance of the best reconstructed graph |
| GSM-N$(\mathcal{G}, \hat{\mathcal{G}})$ | Similarity between N-hop neighborhoods of $\mathcal{G}$ and $\hat{\mathcal{G}}$ | $\mathcal{G}_{best}$ | The best reconstructed graph. |

## A.2 DEFERRED PROOFS

### A.2.1 SPAN CHECK PROOF

Here we show the proof of Thm. 5.1, which we restate here for convenience:

**Theorem 5.1.** *If $n < d$, $\boldsymbol{X}^l_i \in \mathrm{colspan}(\frac{\partial \mathcal{L}}{\partial \boldsymbol{W}^l})$ if and only if $\frac{\partial \mathcal{L}}{\partial \boldsymbol{Y}^l_i} \notin \mathrm{rowspan}(\frac{\partial \hat{\mathcal{L}}}{\partial \boldsymbol{Y}^l_i})$, where $\frac{\partial \hat{\mathcal{L}}}{\partial \boldsymbol{Y}^l_i}$ denotes the matrix $\frac{\partial \mathcal{L}}{\partial \boldsymbol{Y}^l}$ with its $i$-th row removed.*

*Proof.* For notational clarity, we omit the layer index $l$ in our proof. We separate the proof in 3 steps:

- Step 1: $\boldsymbol{X}^T_i \in \mathrm{colspan}(\frac{\partial \mathcal{L}}{\partial \boldsymbol{W}})$ if and only if there is a vector $\alpha_i$, such that $\frac{\partial \mathcal{L}}{\partial \boldsymbol{Y}} \alpha_i = e_i$, where $e_i$ is the $i$-th standard basis vector.

- Step 2: There is a vector $\alpha_i$, such that $\frac{\partial \mathcal{L}}{\partial \boldsymbol{Y}} \alpha_i = e_i$ if and only if $\mathrm{null}(\frac{\partial \hat{\mathcal{L}}}{\partial \boldsymbol{Y}_i}) \not\subseteq \mathrm{null}(\frac{\partial \mathcal{L}}{\partial \boldsymbol{Y}_i})$.

- Step 3: $\mathrm{null}(\frac{\partial \hat{\mathcal{L}}}{\partial \boldsymbol{Y}_i}) \not\subseteq \mathrm{null}(\frac{\partial \mathcal{L}}{\partial \boldsymbol{Y}_i})$ is equivalent to $\frac{\partial \mathcal{L}}{\partial \boldsymbol{Y}_i} \notin \mathrm{rowspan}(\frac{\partial \hat{\mathcal{L}}}{\partial \boldsymbol{Y}_i})$.

**Step 1 ($\boldsymbol{X}^T_i \in \mathrm{colspan}(\frac{\partial \mathcal{L}}{\partial \boldsymbol{W}}) \iff \exists \alpha_i . \frac{\partial \mathcal{L}}{\partial \boldsymbol{Y}} \alpha_i = e_i$):**

($\Rightarrow$) First, we begin by expressing $\frac{\partial \mathcal{L}}{\partial \boldsymbol{W}}$ using the following common result from Petrov et al. (2024):

$$\frac{\partial \mathcal{L}}{\partial \boldsymbol{W}} = \boldsymbol{X}^T \frac{\partial \mathcal{L}}{\partial \boldsymbol{Y}},$$

implying that $\boldsymbol{X}_i^T \in \text{colspan}(\boldsymbol{X}^T \frac{\partial \mathcal{L}}{\partial \boldsymbol{Y}}) = \text{rowspan}(\frac{\partial \mathcal{L}}{\partial \boldsymbol{Y}}^T \boldsymbol{X})$. This can be rewritten as $\exists \alpha_i. \alpha_i^T \frac{\partial \mathcal{L}}{\partial \boldsymbol{Y}}^T \boldsymbol{X} = \boldsymbol{X}_i^T$. Assuming $\boldsymbol{X} \in \mathbb{R}^{n \times d}$ is full-rank, then there exists a right-inverse $\boldsymbol{X}^{-R}$, as $\text{rank}(\boldsymbol{X}) = n < d$.

$$\alpha_i^T \frac{\partial \mathcal{L}}{\partial \boldsymbol{Y}}^T \boldsymbol{X} \boldsymbol{X}^{-R} = \boldsymbol{X}_i^T \boldsymbol{X}^{-R} \Rightarrow \alpha_i^T \frac{\partial \mathcal{L}}{\partial \boldsymbol{Y}}^T = e_i^T \iff \frac{\partial \mathcal{L}}{\partial \boldsymbol{Y}} \alpha_i = e_i$$

It is notable that $\boldsymbol{X}$ not being full-rank still allows for all nodes with feature vectors in $\boldsymbol{X}$ to pass the span check, however it is possible that some hallucinated inputs might also pass the check.

($\Leftarrow$)

$$\frac{\partial \mathcal{L}}{\partial \boldsymbol{Y}} \alpha_i = e_i \iff$$
$$\alpha_i^T \frac{\partial \mathcal{L}}{\partial \boldsymbol{Y}}^T = e_i^T \iff$$
$$\alpha_i^T \frac{\partial \mathcal{L}}{\partial \boldsymbol{Y}}^T X = X_i^T$$

Thus $\boldsymbol{X}_i^T \in \text{rowspan}(\frac{\partial \mathcal{L}}{\partial \boldsymbol{Y}}^T \boldsymbol{X}) = \text{colspan}(\boldsymbol{X}^T \frac{\partial \mathcal{L}}{\partial \boldsymbol{Y}}) = \text{colspan}(\frac{\partial \mathcal{L}}{\partial \boldsymbol{W}})$. Therefore $\boldsymbol{X}_i^T \in \text{colspan}(\frac{\partial \mathcal{L}}{\partial \boldsymbol{W}})$.

**Step 2** ($\exists \alpha_i . \frac{\partial \mathcal{L}}{\partial \boldsymbol{Y}} \alpha_i = e_i \iff \text{null}(\frac{\hat{\partial} \mathcal{L}}{\partial \boldsymbol{Y}_i}) \not\subseteq \text{null}(\frac{\partial \mathcal{L}}{\partial \boldsymbol{Y}_i})$): First, for both directions of the proof, we can separate $\frac{\partial \mathcal{L}}{\partial \boldsymbol{Y}} \alpha_i = e_i$ into 2 different requirements:

$$\frac{\hat{\partial} \mathcal{L}}{\partial \boldsymbol{Y}_i} \alpha_i = \boldsymbol{0} \tag{3}$$

$$\frac{\partial \mathcal{L}}{\partial \boldsymbol{Y}_i} \alpha_i = 1 \tag{4}$$

($\Rightarrow$) Assuming the existence of $\alpha_i$ with $\frac{\partial \mathcal{L}}{\partial \boldsymbol{Y}_i} \alpha_i = e_i$, we know that Eq. 3 and Eq. 4 hold. It is evident with $\alpha_i \in \text{null}(\frac{\hat{\partial} \mathcal{L}}{\partial \boldsymbol{Y}_i})$ but $\alpha_i \notin \text{null}(\frac{\partial \mathcal{L}}{\partial \boldsymbol{Y}_i})$ that $\text{null}(\frac{\hat{\partial} \mathcal{L}}{\partial \boldsymbol{Y}_i}) \not\subseteq \text{null}(\frac{\partial \mathcal{L}}{\partial \boldsymbol{Y}_i})$.

($\Leftarrow$) First of all, we note that $\frac{\partial \mathcal{L}}{\partial \boldsymbol{Y}_i} \in \mathbb{R}^{1 \times d}$ has rank 1, as $\frac{\partial \mathcal{L}}{\partial \boldsymbol{Y}_i}$ contains a non-zero entry under normal training conditions. Therefore, $\frac{\partial \mathcal{L}}{\partial \boldsymbol{Y}_i}$ has $\text{nullity}(\frac{\partial \mathcal{L}}{\partial \boldsymbol{Y}_i}) = n - 1$ due to the rank-nullity theorem. $\text{null}(\frac{\hat{\partial} \mathcal{L}}{\partial \boldsymbol{Y}_i}) \not\subseteq \text{null}(\frac{\partial \mathcal{L}}{\partial \boldsymbol{Y}_i})$ implies that there exists an $\alpha_i' \in \text{null}(\frac{\hat{\partial} \mathcal{L}}{\partial \boldsymbol{Y}_i})$, such that $\alpha_i' \notin \text{null}(\frac{\partial \mathcal{L}}{\partial \boldsymbol{Y}_i})$ (since $\text{nullity}(\frac{\partial \mathcal{L}}{\partial \boldsymbol{Y}_i}) = n - 1 < n$ this set is non-empty). For that $\alpha_i$, the following hold:

$$\frac{\hat{\partial} \mathcal{L}}{\partial \boldsymbol{Y}_i} \alpha_i' = \boldsymbol{0}$$
$$\frac{\partial \mathcal{L}}{\partial \boldsymbol{Y}_i} \alpha_i' = c$$

Therefore, if we take $\alpha_i = \frac{1}{c} \alpha_i'$, $\alpha_i$ would satisfy both (2) and (3), giving us a valid solution.

**Step 3:** ($\text{null}(\frac{\hat{\partial} \mathcal{L}}{\partial \boldsymbol{Y}_i}) \not\subseteq \text{null}(\frac{\partial \mathcal{L}}{\partial \boldsymbol{Y}_i}) \iff \frac{\partial \mathcal{L}}{\partial \boldsymbol{Y}_i} \notin \text{rowspan}(\frac{\hat{\partial} \mathcal{L}}{\partial \boldsymbol{Y}_i})$) First of all, the statement is equivalent to negating both sides, or $\text{null}(\frac{\hat{\partial} \mathcal{L}}{\partial \boldsymbol{Y}_i}) \subseteq \text{null}(\frac{\partial \mathcal{L}}{\partial \boldsymbol{Y}_i}) \iff \frac{\partial \mathcal{L}}{\partial \boldsymbol{Y}_i} \in \text{rowspan}(\frac{\hat{\partial} \mathcal{L}}{\partial \boldsymbol{Y}_i})$, which can be shown by the following steps:

$$\text{null}(\frac{\hat{\partial} \mathcal{L}}{\partial \boldsymbol{Y}_i}) \subseteq \text{null}(\frac{\partial \mathcal{L}}{\partial \boldsymbol{Y}_i}) \iff \text{null}(\frac{\partial \mathcal{L}}{\partial \boldsymbol{Y}_i})^C \subseteq \text{null}(\frac{\hat{\partial} \mathcal{L}}{\partial \boldsymbol{Y}_i})^C$$
$$\iff \text{rowspan}(\frac{\partial \mathcal{L}}{\partial \boldsymbol{Y}_i}) \subseteq \text{rowspan}(\frac{\hat{\partial} \mathcal{L}}{\partial \boldsymbol{Y}_i})$$
$$\iff \frac{\partial \mathcal{L}}{\partial \boldsymbol{Y}_i} \in \text{rowspan}(\frac{\hat{\partial} \mathcal{L}}{\partial \boldsymbol{Y}_i})$$

Here we used that the complenetary subspace of the null space of matrix is the rowspan of the matrix $\text{null}(M)^C = \text{rowspan}(M)$. The last step follows from that the fact that $\frac{\partial \mathcal{L}}{\partial \boldsymbol{Y}_i}$ is a single common vector, and therefore all vectors in $\text{colspan}(\frac{\partial \mathcal{L}}{\partial \boldsymbol{Y}_i})$ are of the form $\lambda \frac{\partial \mathcal{L}}{\partial \boldsymbol{Y}_i}$. This concludes our proof.

$\square$

We extend this by presenting the proof for Cor. 5.2:

**Corollary 5.2.** *For $\frac{\partial \mathcal{L}}{\partial \boldsymbol{Z}^l}$ of full-rank, $n < d$, if the (possibly normalized) adjacency matrix at layer $l$, $\boldsymbol{A} \in \mathbb{R}^{n \times n}$, $\boldsymbol{X}_i^l \in \text{colspan}(\frac{\partial \mathcal{L}}{\partial \boldsymbol{W}^l})$ if and only if $\boldsymbol{A}_i^T \notin \text{colspan}(\hat{\boldsymbol{A}}_i)$. Further, if $\boldsymbol{A}$ is full-rank, then $\boldsymbol{X}_i^l \in \text{colspan}(\frac{\partial \mathcal{L}}{\partial \boldsymbol{W}^l})$ for all $i = 1, 2, \ldots, n$.*

*Proof.* For notational clarity, we omit the layer index $l$ in our proof.

From $\boldsymbol{Z} = \boldsymbol{AY}$, by applying the following common result from Petrov et al. (2024) (replacing $\boldsymbol{W}$ with $\boldsymbol{Y}$, and $\boldsymbol{Y}$ with $\boldsymbol{Z}$), we obtain:

$$\frac{\partial \mathcal{L}}{\partial \boldsymbol{Y}} = \boldsymbol{A}^T \frac{\partial \mathcal{L}}{\partial \boldsymbol{Z}}$$

Therefore, $\frac{\partial \mathcal{L}}{\partial \boldsymbol{Y}_i} = \boldsymbol{A}_i^T \frac{\partial \mathcal{L}}{\partial \boldsymbol{Z}}$. We will now prove that if $\frac{\partial \mathcal{L}}{\partial \boldsymbol{Z}}$ is full-rank, then $\boldsymbol{A}_i^T \notin \text{colspan}(\hat{\boldsymbol{A}}_i) \iff \frac{\partial \mathcal{L}}{\partial \boldsymbol{Y}_i} \notin \text{rowspan}(\frac{\hat{\partial \mathcal{L}}}{\partial \boldsymbol{Y}_i})$. This is equivalent to proving the converse, or $\boldsymbol{A}_i^T \in \text{colspan}(\hat{\boldsymbol{A}}_i) \iff \frac{\partial \mathcal{L}}{\partial \boldsymbol{Y}_i} \in \text{rowspan}(\frac{\hat{\partial \mathcal{L}}}{\partial \boldsymbol{Y}_i})$

$(\Rightarrow)$ $\boldsymbol{A}_i^T \in \text{colspan}(\hat{\boldsymbol{A}}_i)$ implies that there exist coefficients $\alpha_1, \alpha_2, \cdots, \alpha_n$, such that:

$$\boldsymbol{A}_i^T = \sum_{j=\{1,2,\ldots,N\}\setminus\{i\}} \alpha_j \boldsymbol{A}_j^T,$$

Multiplying both sides by $\frac{\partial \mathcal{L}}{\partial \boldsymbol{Z}}$ gives:

$$\boldsymbol{A}_i^T \frac{\partial \mathcal{L}}{\partial \boldsymbol{Z}} = \sum_{j=\{1,2,\ldots,N\}\setminus\{i\}} \alpha_j \boldsymbol{A}_j^T \frac{\partial \mathcal{L}}{\partial \boldsymbol{Z}} \iff \frac{\partial \mathcal{L}}{\partial \boldsymbol{Y}_i} = \sum_{j=\{1,2,\ldots,N\}\setminus\{i\}} \alpha_j \frac{\partial \mathcal{L}}{\partial \boldsymbol{Y}_j} \iff \frac{\partial \mathcal{L}}{\partial \boldsymbol{Y}_i} \in \text{rowspan}(\frac{\hat{\partial \mathcal{L}}}{\partial \boldsymbol{Y}_i})$$

$(\Leftarrow)$ We can similarly rewrite $\frac{\partial \mathcal{L}}{\partial \boldsymbol{Y}_i} \in \text{rowspan}(\frac{\hat{\partial \mathcal{L}}}{\partial \boldsymbol{Y}_i})$ as:

$$\frac{\partial \mathcal{L}}{\partial \boldsymbol{Y}_i} = \sum_{j=\{1,2,\ldots,N\}\setminus\{i\}} \alpha_j \frac{\partial \mathcal{L}}{\partial \boldsymbol{Y}_j} \iff \boldsymbol{A}_i^T \frac{\partial \mathcal{L}}{\partial \boldsymbol{Z}} = \sum_{j=\{1,2,\ldots,N\}\setminus\{i\}} \alpha_j \boldsymbol{A}_j^T \frac{\partial \mathcal{L}}{\partial \boldsymbol{Z}}$$

As $\frac{\partial \mathcal{L}}{\partial \boldsymbol{Z}}$ is full-rank, then there exists a right-inverse $\frac{\partial \mathcal{L}}{\partial \boldsymbol{Z}}^{-R}$. Multiplying on both sides gives:

$$\boldsymbol{A}_i^T \frac{\partial \mathcal{L}}{\partial \boldsymbol{Z}} \frac{\partial \mathcal{L}}{\partial \boldsymbol{Z}}^{-R} = \sum_{j=\{1,2,\ldots,N\}\setminus\{i\}} \alpha_j \boldsymbol{A}_j^T \frac{\partial \mathcal{L}}{\partial \boldsymbol{Z}} \frac{\partial \mathcal{L}}{\partial \boldsymbol{Z}}^{-R} \iff$$

$$\boldsymbol{A}_i^T = \sum_{j=\{1,2,\ldots,N\}\setminus\{i\}} \alpha_j \boldsymbol{A}_j^T \iff \boldsymbol{A}_i^T \in \text{colspan}(\hat{\boldsymbol{A}}_i)$$

From here, we can now apply Thm. 5.1, which gives us that $\boldsymbol{X}_i \in \text{colspan}(\frac{\partial \mathcal{L}}{\partial \boldsymbol{W}})$ if and only if $\boldsymbol{A}_i^T \notin \text{colspan}(\hat{\boldsymbol{A}}_i)$. If $\boldsymbol{A}$ is full-rank, then $\boldsymbol{A}_i^T \notin \text{colspan}(\hat{\boldsymbol{A}}_i)$ for all $i = 1, 2, \cdots, n$, implying that $\boldsymbol{X}_i \in \text{colspan}(\frac{\partial \mathcal{L}}{\partial \boldsymbol{W}})$ for all $i$. This concludes the proof. $\square$

### A.2.2 EMBEDDING RECOVERY PROOF

In this section, first we formally state Asm. A.1 on which Thm. 5.3 is based. Next, using it we provide a proof for Thm. 5.3. Finally, we prove that Asm. A.1 is satisfied for common GNN architectures, such as GCNs and GATs.

For simplicity, in the rest of the section we will denote the node of $\mathcal{G}$ corresponding to the the $i^{\text{th}}$ row of the embedding matrices $\boldsymbol{X}_i^l$ with $v_i$.

**Assumption A.1.** The output corresponding to the node $v_i$ of the $l$-th layer of the GNN, $X_i^{l+1}$, is independent on $X_j^l$ for $\forall v_j \notin \mathcal{N}_{\mathcal{G}}^1(v_i)$.

The assumption intuitively states that for most popular GNNs the embeddings of a node only depends on the embeddings of its neighbourhood on the previous layer. Using this, we now present the proof of Thm. 5.3, restating it for convinience:

**Theorem 5.3.** *For GNNs satisfying Asm. A.1, propagating a correctly reconstructed building block $\mathcal{G}_v^l$ centred at $v$ through the first $l$ GNN layers recovers the original embedding of $v$ at layer $l$:*

$$f_{l-1}(\mathcal{G}_v^l)[j] = X_i^l,$$

*where $i$ is the index of $v$ in the adjacency matrix of the original input graph $\mathcal{G}$ and $j$ is the index of $v$ in the adjacency matrix of $\mathcal{G}_v^l$.*

*Proof.* First we will show that the embedding $\boldsymbol{X}_i^l$ is independent of the embeddings of any nodes that do not belong in the $l$-hop neighborhood of $v$. For $l = 1$ this follows immediately from Asm. A.1. For $l = 2$, we know that $\boldsymbol{X}_i^2$ is only dependent on the embeddings $\boldsymbol{X}_k^1$ of the neighboring nodes $v_k \in \mathcal{N}_{\mathcal{G}}^1(v)$ by applying Asm. A.1. However, the embeddings $\boldsymbol{X}_k^1$ themselves only depends on the embeddings of the nodes in the 1-hop neighborhood of $v_k$. Therefore, the only nodes that can influence $\boldsymbol{X}_i^2$ are $v$, its neighbors, and the neighbors of its neighbors, which exactly comprise the 2-hop neighborhood of $v$. Similarly, we can show that $\boldsymbol{X}_i^l$ is independent of the embeddings of any nodes that are not part of the $l$-hop neighborhood of $v$ by induction.

We now take the full graph $\mathcal{G}$ and remove all edges starting or ending at nodes not in $\mathcal{G}_v^l$, and set their feature values to 0, obtaining $\hat{\mathcal{G}}$. As shown above, this implies the output of $f_{l-1}$ for $\mathcal{G}$ at $v$ is the same as the one for $\hat{\mathcal{G}}$. Note that applying the network on $\hat{\mathcal{G}}$ is the same as applying it to $\mathcal{G}_v^l$ except for few the additional embeddings produced for the disconnected nodes. By reindexing we arrive at our formula.

$\square$

Finally, we prove our assumption Asm. A.1 holds for all GNN architectures considered in this paper:

**Theorem A.2.** *Asm. A.1 holds for GCNs*

*Proof.* For GCN models, the formula for computing the $i^{\text{th}}$ embedding at layer $l + 1$, $\boldsymbol{X}_i^{l+1}$, can be expressed as:

$$\boldsymbol{X}_i^{l+1} = \sigma\left(\sum_{v_j \in \mathcal{N}_{\mathcal{G}}^1(v_i)} \boldsymbol{A}_{i,j}^l \mathbf{W}^l \boldsymbol{X}_j^l\right)$$

where $\boldsymbol{A}_{i,j}^l$ is the normalized strength of the connection between $v_i$ and $v_j$ at layer $l$ given by:

$$\boldsymbol{A}_{i,j}^l = \frac{1}{\sqrt{deg(v_i)deg(v_j)}}.$$

Assuming the degrees of the nodes $v_j \in \mathcal{N}_{\mathcal{G}}^1(v_i)$ are known, $\boldsymbol{X}_i^{l+1}$ is independent on $\boldsymbol{X}_j^l$ for $v_j \notin \mathcal{N}_{\mathcal{G}}^1(v_i)$. We note that for a correctly recovered $l$-hop neighbourhood around $v$, the degrees of all nodes in the $(l-1)$-hop neighbourhood around $v$ can be recovered exactly, while for the remaining nodes one can provide a sound lower bound for their degree. Assuming a sound guess on the upper bound on the largest degree in the graph, this allows to recover the degrees of the remaining nodes via enumeration starting from these computed lower bounds. Further, if the input features $\boldsymbol{X}_i^0$ of $v_i$ contain information about the node degree of $v_i$, this information can be easily incorporated to reduce the computational overhead of the enumeration. $\square$

**Theorem A.3.** *Asm. A.1 holds for GATs*

*Proof.* For GAT models, the formula for computing the $i^{\text{th}}$ embedding at layer $l + 1$, $\boldsymbol{X}_i^{l+1}$, can similarly be expressed as:

$$\boldsymbol{X}_i^{l+1} = \sigma\left(\sum_{v_j \in \mathcal{N}_{\mathcal{G}}^1(v_i)} \boldsymbol{A}_{i,j}^l \mathbf{W}^l \boldsymbol{X}_j^l\right)$$

where $\boldsymbol{A}^l_{i,j}$ is the attention coefficient between $v_i$ and $v_j$ at layer $l$ given by:

$$\boldsymbol{A}^l_{i,j} = \frac{\exp\left(\text{LeakyReLU}\left((\mathbf{a}^l)^T[\mathbf{W}^l\boldsymbol{X}^l_i \parallel \mathbf{W}^l\boldsymbol{X}^l_j]\right)\right)}{\sum_{v_k \in \mathcal{N}^1_{\mathcal{G}}(v_i)} \exp\left(\text{LeakyReLU}\left((\mathbf{a}^l)^T[\mathbf{W}^l\boldsymbol{X}^l_i \parallel \mathbf{W}^l\boldsymbol{X}^l_k]\right)\right)}$$

and $a^l$ is a vector of attention parameters.

As $\boldsymbol{X}^{l+1}_i$ only depends on $\boldsymbol{W}^l, \boldsymbol{A}^l_{i,j}$ and $\boldsymbol{X}^l_j$ to be calculated for $v_j \in \mathcal{N}^1_{\mathcal{G}}(v_i)$, and $\boldsymbol{A}^l_{i,j}$ itself also only depends on $\boldsymbol{X}^l_j$ and $\boldsymbol{W}^l$, we show that $\boldsymbol{X}^{l+1}_i$ is independent on $\boldsymbol{X}^l_j$ for $v_j \notin \mathcal{N}^1_{\mathcal{G}}(v_i)$. □

### A.3 GLUING ALGORITHM

In order to establish the gluing operation, we need to first define how we determine which nodes in $\mathcal{G}$ and $\mathcal{G}^B$ can potentially match, so that the building block is glued correctly. This can be achieved recursively, by ensuring that:

- For every pair of matched nodes, their features must be exactly equal.

- The center of $\mathcal{G}^B$ - $c^B$ matches the center of attaching $c$.

- For a node $v \in V$ of $\mathcal{G}$ and its match $v_B \in V^B$, every one of its neighbors $v'$ $((v, v') \in E)$ has to match a neighbor $v'_B$ of $v_B$ $((v_B, v'_B) \in E^B)$.

This recursive definition can be satisfied by exploring the possible matchings from the center $c$ outwards. In particular, we explore the $i$-hop neighborhoods one by one for $i = 1, 2, \cdots, l$. We try all possible matchings $v_B$ for the given node $v$, such that their features match, and the matches of its neighbors (in particular those which have already been traversed) are also connected to $v_B$ (Line 7).

After all possible matchings are generated, it is easy to substitute any nodes in the original graph with those in the building block, in order to obtain the set of all possible gluings (Lines 13–16)

---

**Algorithm 5** Gluing a graph with a building block

```
 1: function GLUE(𝒢, 𝒢ᴮ, c, l )
 2:     M̂ ← {{(c, cᴮ)}}
 3:     for i = 1, 2, ⋯, l do                          ▷ Generate all correct matchings
 4:         for v ∈ {v|dist(c, v) = i} do
 5:             M̂_new ← {}
 6:             for ℳ ∈ M̂ do
 7:                 M̂_new ← M̂_new × {(v, v_B)|𝐗_v = 𝐗_{v_B} ∧ ∩_{(v,v')∈E,(v',v'_B)∈ℳ}(v_B, v'_B) ∈ Eᴮ}
 8:             M̂ ← M̂_new
 9:     𝔾 ← {}
10:     for ℳ ∈ M̂ do                                  ▷ Transform matchings into valid graphs
11:         V̂ ← V \ 𝒩^l_𝒢(c) ∪ V_B
12:         Ê ← Eᴮ
13:         for (v, v') ∈ E do
14:             v = v_B if ∃v_B.(v, v_B) ∈ ℳ else v
15:             v' = v'_B if ∃v'_B.(v', v'_B) ∈ ℳ else v'
16:             Ê ← Ê ∪ {(v, v')}
17:         𝔾 ← 𝔾 ∪ {(V̂, Ê)}
18:     return 𝔾
```

---

## A.4 FEATURE-BY-FEATURE FILTERING ALGORITHM

---

**Algorithm 6** Filtering nodes feature-by-feature

---

1: **function** FILTERNODES($\{\mathcal{F}_i\}_{i\in\{1,\cdots,m\}}, \frac{\partial\mathcal{L}}{\partial\mathbf{W}_0}, \tau$)
2: $\quad \{\mathcal{T}_{0,i}^*\}_{i\in\{1,\cdots,m\}}, d_{sum} \leftarrow \{\emptyset\}_{i\in\{1,\cdots,m\}}, |\mathcal{F}_1|$
3: $\quad$ **for** $k \in \{1,\cdots,m\}$ **do**
4: $\qquad \mathcal{T}_{0,k}^* \leftarrow \mathcal{T}_{0,k-1}^* \times \mathcal{F}_k$
5: $\qquad d_{sum} \leftarrow d_{sum} + |\mathcal{F}_k|$
6: $\qquad$ **if** $d_{sum} > \text{rank}(\frac{\partial\mathcal{L}}{\partial\mathbf{W}_0})$ **then** $\qquad\qquad \triangleright$ This is a requirement for the filtering to work
7: $\qquad\qquad \mathcal{T}_{0,k}^* \leftarrow \text{FILTER}(\mathcal{T}_{0,k}^*, \frac{\partial\mathcal{L}}{\partial\mathbf{W}_0}[: d_{sum}], \tau, \lambda v.\mathbf{X}_v^0)$
8: $\quad$ **return** $\mathcal{T}_{0,m}^*$

---

Here we describe how we build and filter 0-hop neighborhoods, i.e. the single node client feature vectors. While DAGER directly enumerates and filters all possible text input feature vectors, for many graph data applications this procedure is impractical. To this end, we instead recover the node features one-by-one, as described next.

For the first feature, we generate all partial input vectors associated with the values in $\mathcal{F}_1$ and then filter them to create the set of consistent partial input vectors $\mathcal{T}_{0,1}^*$. To do so, we apply our span-check on the truncated input layer gradients $\frac{\partial\mathcal{L}}{\partial W^0}[: |\mathcal{F}_1|]$ corresponding to the input entries of the first feature. This is possible since, by Thm. 5.1, the truncated $i$-th row of $\mathbf{X}$, $\mathbf{X}[i, : |\mathcal{F}_1|]$, is in $\text{colspan}(\frac{\partial\mathcal{L}}{\partial W^0}[: |\mathcal{F}_1|])$. For each subsequent feature, we similarly filter the set of vectors corresponding to combinations between a partially recovered vector in $\mathcal{T}_{0,i-1}^*$ and a value in $\mathcal{F}_i$ by applying Thm. 5.1 to obtain the set of consistent partial input vectors up to feature $i$ $\mathcal{T}_{0,i}^*$. Finally, we obtain $\mathcal{T}_0^* = \mathcal{T}_{0,m}^*$. A pseudocode for this approach is provided in Alg. 6.

## A.5 STRUCTURE-BASED FILTERING ALGORITHM

---

**Algorithm 7** Structure filtering algorithm

---

1: **function** STRUCTUREFILTER($\mathcal{T}_L^*, \frac{\partial\mathcal{L}}{\partial\mathbf{W}}$)
2: $\quad$ **for** $\mathcal{G} \in \mathcal{T}_L^*$ **do**
3: $\qquad$ **if** $\Delta_\mathcal{G} == 0$ **then**
4: $\qquad\qquad$ **return** $\mathcal{G}$
5: $\qquad CanGlue \leftarrow True$
6: $\qquad$ **for** $v \in V$ **do**
7: $\qquad\qquad$ **if** $!\exists\mathcal{G}^B \in \mathcal{T}_L^*.\text{glue}(\mathcal{G}, \mathcal{G}^B, v)$ **then**
8: $\qquad\qquad\qquad CanGlue \leftarrow False$
9: $\qquad\qquad\qquad$ **break**
10: $\qquad$ **if** $CanGlue$ **then**
11: $\qquad\qquad \mathcal{T}_B^* \leftarrow \mathcal{T}_B^* \cup \{\mathcal{G}\}$
12: $\quad$ **return** $\mathcal{T}_B^*$

---

## A.6 DEPTH-FIRST SEARCH IMPLEMENTATION

We present the pseudocode for branching the DFS tree:

**Cycles exploration** After every step of gluing possible building blocks from $\mathcal{T}_B^*$ to $\mathcal{G}_{\text{curr}}$ at node $v$ (line 5 in Alg. 8), we enumerate all sets $S$ of pairs ($v_1 \in \mathcal{G}_{\text{curr}}, v_2 \notin \mathcal{G}_{\text{curr}}$) of vertices of any graph $\mathcal{G}' \in \mathbb{G}$ such that the features of $v_1$ and $v_2$ match (line 9 in Alg. 8). For every such set $S$, we additionally consider for exploration the graph $\hat{\mathcal{G}}' = \text{overlap}(\mathcal{G}', S)$ created by overlapping each pair of vertices in $S$ (line 10 in Alg. 8). Intuitively, this allows us to reconstruct any cycle of the input graph. All valid overlaps, including those with $S = \emptyset$, where the graph remains unchanged, are then added as branches of the search space (line 8 in Alg. 4).

---

**Algorithm 8** Generate graphs for branch

---

1: **function** BRANCH($\mathcal{T}_B^*$, $\mathcal{G}_{\text{curr}}$, $v$)
2:     $\mathbb{G}_{\text{new}} \leftarrow \{\}$
3:
4:     **for** $\mathcal{B} \in \mathcal{T}_B^*$ **do**
5:         $\mathbb{G} \leftarrow \text{glue}(\mathcal{G}_{\text{curr}}, \mathcal{B}, v)$
6:         **if** $\mathbb{G} \neq \emptyset$ **then**
7:             $\mathbb{G}_{\text{new}} \leftarrow \mathbb{G}_{\text{new}} \cup \mathbb{G}$
8:             **for** $\mathcal{G}' \in \mathbb{G}$ **do**
9:                 **for** $S \subseteq \{V(\mathcal{G}') \setminus V(\mathcal{G}_{\text{curr}})\} \times V(\mathcal{G}_{\text{curr}})$ **do**
10:                     $\mathbb{G}_{\text{new}} \leftarrow \mathbb{G}_{\text{new}} \cup \{\text{overlap}(\mathcal{G}', S)\}$
11:
12:     **return** $\mathbb{G}_{\text{new}}$

---

**Node overlaping** In this paragraph, we describe how the overlapping function (used in line 10 in Alg. 8) works. Given a graph $\mathcal{G}'$ and a set of pairs of nodes $S$, if some pair has two nodes with different node feature vectors, the function returns an empty graph. Otherwise, for each pair $(v_1, v_2) \in S$ we modify the current graph by deleting $v_2$ and connect add an edge between $v_1$ and any node that was previously a neighbor of $v_2$ but not $v_1$. After doing these overlaps for every pair of nodes in $S$, the new graph is returned.

## B THE GSM METRIC

We designed the GSM set of metrics so that they satisfy the following three qualities:

- The metric should be efficiently computable in polynomial time
- It should capture both structural and feature-wise information
- Isomorphic graphs should be guaranteed to achieve a 100% score

To this end, we define GSM-N$_F(\mathcal{G}, \hat{\mathcal{G}})$ under a set of functions $F = \{F_k\}_{k=1}^N$, where for all $k$ $F_k : \mathcal{G} \to \mathbb{R}^{|\mathcal{V}| \times d}$ is a function that aggregates the feature vectors for each $k$-hop neighborhood. This allows us to measure the similarites in features across increasingly larger subgraphs, which capture the structure around each node. In our case we utilise a randomly initialised $\geq k$-layer GCN to achieve such a mapping.

We note that a precise evaluation of the metric requires for us to match the 2 graphs as accurately as possible. Since exact matching of graphs has no known polynomial-time algorithm (Babai, 2016), we match the graph nodes by applying the Hungarian matching algorithm (Frank, 2005) for minimizing a cost function $C$ that captures the feature difference across (0-5)-hop neighborhoods:

$$C_{ij} = \sum_{k=0}^{2} \sum_{m=1}^{d} (F_k(\mathcal{G}) - F_k(\hat{\mathcal{G}}))_m^2$$

We can hence define:

$$\text{GSM-N}_F(\mathcal{G}, \hat{\mathcal{G}}) = \begin{cases} \text{F1-Score}(F_N(\mathcal{G}), F_N(\hat{\mathcal{G}})) & \text{if } F_N(\mathcal{G}) \text{ - discrete} \\ R^2(F_N(\mathcal{G}), F_N(\hat{\mathcal{G}})) & \text{if } F_N(\mathcal{G}) \text{ - continuous} \end{cases}$$

First of all, the NP-complete nature of the subgraph isomorphism problem makes it difficult to do any subgraph or full-graph matching, which we tackle by utilising the hidden states of a GCN to create an approximate matching between nodes. The method described above ensures that for isomorphic graphs, the correct k-hop neighborhoods will be matched and the GSM-k metric will achieve a perfect score.

The second requirement is rarely satisfied by metrics defined in the literature (i.e. the edit distance), as comparison studies on coloured graphs are limited. Our solution to these problems was inspired by the ROUGE set of metrics (Lin, 2004), used for evaluation of textual similarity. Instead of

comparing sequences such as unigrams or bigrams like ROUGE, we instead compute continuous properties of graphs on the scale of different k-hop neighbourhoods. This rationale allows us to compare both node-based and structural properties using a simple methodology.

## C   ADDITIONAL EXPERIMENTS

Here we present additional experiments that are not part of the main text.

### C.1   HUMAN EVALUATION FOR THE GSM SET OF METRICS

We performed a human evaluation, where 3 experts in Graph Theory and Chemistry were shown 120 sample reconstructions of molecules, as given by DLG and GRAIN. The samples were shuffled, and the participants were tasked to assign a score from 0 to 10, with the following instructions:

"Thank you for agreeing to participate in this study on the quality of graph reconstructions! We have gathered a set of graphs, coupled with the best-effort reconstruction. Please give each pair a score of 0-10, where 0 is a complete lack of similarity, and 10 is a perfect match. When assigning a score, take into account the *structure* of the two graphs, as well as the *atom type* for matching atoms, and also be wary that 2 graphs might be *isomorphic*, but have different pictures. Please disregard the connections between atoms, as the methods we used do not recover any edge properties. Give your, as best as possible, score on how similar the graphs are with respect to these properties."

Reconstructed samples from both GRAIN and DLG were shuffled and anonymized before being presented to the participants. We report the average scores for each algorithm, multiplied by a factor of 10 to match the order of magnitude of the GRAPH metrics, and present the results in Tab. 1. We observe very good correlation between our metrics and the reported human scores, even though our metrics are slightly more lenient to completely wrong reconstructions, compared to the evaluators. This leniency provides a slight advantage to the baseline attacks when measured using our metrics, as the baselines fail catastrophically more often.

Based on these studies, we also show in Tab. 7 that our partial reconstructions are deemed more significant than what the metric suggests, likely meaning that there are examples which present significant information leakage. In contrast, high-scoring examples from the DLG attacks have been rated as essentially uninformative.

Table 7: Score discrepancy examples between human evaluators and the GRAPH set of metrics. G-1 stands for the GRAPH-1 metric.

| GT | GRAIN | G-1 | Study | GT | DLG | G-1 | Study |
|----|-------|-----|-------|----|-----|-----|-------|
| | | 62.0 | 93.3 | | | 52.7 | 10.0 |
| | | 41.3 | 63.3 | | | 61.0 | 23.3 |
| | | 33.5 | 56.7 | | | 41.0 | 0.0 |

Additional examples of molecule reconstructions comparing GRAIN, DLG, and TabLeak are shown in Fig. 5. In this set of examples, the first 3 columns show the exact reconstruction of the input. We also highlight that in cases where GRAIN does not managed to recover the entire graph, the attack can reconstruct subgraphs of the input (4th column), and a more realistic approximation otherwise (5th column).

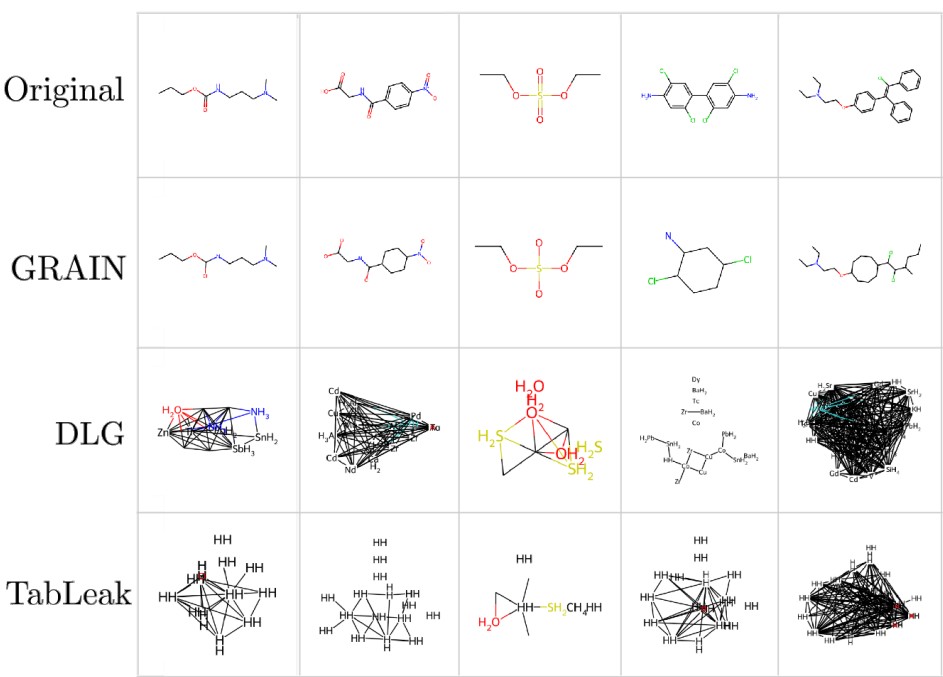

Figure 5: Examples of molecule reconstructions compared between GRAIN, DLG, and TabLeak.

## C.2 ADDITIONAL CHEMICAL DATASETS

Table 8: Results (in %) of main experiments on 3 biochemical datasets – Tox21, Clintox, BBBP. Here "$+A$" refers to the baseline attack with the input adjacency matrix given.

|  |  | GSM-0 | GSM-1 | GSM-2 | FULL | Runtime[h] |
|---|---|---|---|---|---|---|
| Tox21 | GRAIN | $86.9^{+4.2}_{-5.7}$ | $83.9^{+5.2}_{-6.9}$ | $82.6^{+5.7}_{-7.4}$ | $68.0 \pm 1.7$ | 14.3 |
|  | DLG | $31.8^{+4.5}_{-4.3}$ | $20.3^{+5.5}_{-4.8}$ | $22.8^{+6.6}_{-5.6}$ | $1.0 \pm 0.2$ | 3.3 |
|  | DLG $+A$ | $54.7^{+3.9}_{-4.2}$ | $60.1^{+4.6}_{-5.2}$ | $76.7^{+3.6}_{-4.8}$ | $1.0 \pm 0.2$ | **3.1** |
|  | TabLeak | $25.1^{+5.1}_{-4.3}$ | $12.4^{+5.5}_{-4.3}$ | $10.8^{+5.6}_{-3.9}$ | $1.0 \pm 0.2$ | 13.1 |
|  | TabLeak $+A$ | $55.6^{+3.9}_{-3.9}$ | $57.7^{+4.1}_{-4.6}$ | $73.8^{+2.8}_{-3.5}$ | $1.0 \pm 0.2$ | 12.3 |
| Clintox | GRAIN | $73.7^{+5.7}_{-6.5}$ | $68.4^{+6.7}_{-7.8}$ | $66.8^{+7.0}_{-7.6}$ | $36.0 \pm 1.2$ | 24.1 |
|  | DLG | $24.0^{+4.1}_{-3.8}$ | $10.3^{+4.8}_{-3.6}$ | $12.2^{+5.5}_{-4.2}$ | $1.0 \pm 0.2$ | 3.5 |
|  | DLG $+A$ | $52.5^{+3.2}_{-3.6}$ | $52.6^{+4.1}_{-4.7}$ | $72.3^{+3.2}_{-3.9}$ | $1.0 \pm 0.2$ | **3.2** |
|  | TabLeak | $17.6^{+3.7}_{-2.8}$ | $6.0^{+4.0}_{-2.4}$ | $5.4^{+4.2}_{-2.5}$ | $1.0 \pm 0.2$ | 15.2 |
|  | TabLeak $+A$ | $54.0^{+3.4}_{-3.3}$ | $52.0^{+3.8}_{-4.2}$ | $62.8^{+3.3}_{-4.2}$ | $1.0 \pm 0.2$ | 14.5 |
| BBBP | GRAIN | $71.7^{+5.9}_{-6.8}$ | $66.8^{+6.9}_{-7.7}$ | $64.9^{+7.2}_{-8.0}$ | $38.0 \pm 1.2$ | 23.7 |
|  | DLG | $22.6^{+3.6}_{-3.3}$ | $8.8^{+4.9}_{-3.2}$ | $10.0^{+5.3}_{-3.7}$ | $0.0 \pm 0.0$ | 3.9 |
|  | DLG $+A$ | $51.6^{+3.1}_{-3.6}$ | $50.1^{+3.8}_{-4.5}$ | $70.6^{+3.1}_{-4.2}$ | $0.0 \pm 0.0$ | **3.1** |
|  | TabLeak | $17.6^{+3.8}_{-2.8}$ | $6.3^{+3.8}_{-2.5}$ | $4.7^{+3.7}_{-2.3}$ | $0.0 \pm 0.0$ | 12.6 |
|  | TabLeak $+A$ | $59.1^{+3.1}_{-3.6}$ | $59.4^{+3.6}_{-4.3}$ | $71.9^{+2.9}_{-4.0}$ | $0.0 \pm 0.0$ | 12.5 |

In this section, we present our results on additional chemical datasets, namely Clintox and BBBP (Wu et al., 2018). We highlight in Tab. 8 that GRAIN generalizes across all settings, retaining

its increased performance over the baseline attacks. We reaffirm that we achieve these results despite running for time comparable to the one of Tableak.

## C.3 Additional Ablation Studies

We perform additional ablation studies on various assumptions and parameters, demonstrating their effects on GRAIN. In particular, we observe how architectural changes might affect our performance, or how properties of the data might influence reconstructability.

Table 9: Results (in %) of main experiments with the LBFGS optimizer. Here "$+A$" refers to the baseline attack with the input adjacency matrix given.

| | | GCN | | | | | GAT | | | | |
|---|---|---|---|---|---|---|---|---|---|---|---|
| | | GSM-0 | GSM-1 | GSM-2 | FULL | Min/Rec | GSM-0 | GSM-1 | GSM-2 | FULL | Min/Rec |
| CiteSeer | GRAIN | $62.5^{+7.7}_{-8.2}$ | $31.0^{+8.0}_{-7.8}$ | $31.6^{+8.1}_{-8.1}$ | $20.0 \pm 0.8$ | 1.5 | $79.3^{+4.7}_{-6.3}$ | $69.1^{+6.1}_{-6.4}$ | $69.6^{+6.2}_{-6.0}$ | $61.0 \pm 1.6$ | 0.8 |
| | DLG | $67.7^{+3.9}_{-3.7}$ | $0.0^{+0.0}_{-0.0}$ | $0.3^{+0.5}_{-0.3}$ | $0.0 \pm 0.0$ | 24.8 | $67.7^{+3.9}_{-3.7}$ | $0.0^{+0.0}_{-0.0}$ | $0.0^{+0.0}_{-0.0}$ | $0.0 \pm 0.0$ | 31.0 |
| | DLG $+A$ | $67.7^{+4.1}_{-3.7}$ | $0.0^{+0.0}_{-0.0}$ | $0.0^{+0.0}_{-0.0}$ | $0.0 \pm 0.0$ | 29.9 | $67.7^{+4.0}_{-3.7}$ | $0.0^{+0.0}_{-0.0}$ | $0.0^{+0.0}_{-0.0}$ | $0.0 \pm 0.0$ | 27.7 |
| | TabLeak | $67.7^{+3.9}_{-3.7}$ | $0.0^{+0.0}_{-0.0}$ | $0.0^{+0.0}_{-0.0}$ | $0.0 \pm 0.0$ | 158.8 | $67.7^{+3.9}_{-3.8}$ | $0.0^{+0.0}_{-0.0}$ | $0.0^{+0.0}_{-0.0}$ | $0.0 \pm 0.0$ | 153.0 |
| | TabLeak $+A$ | $67.7^{+4.0}_{-3.7}$ | $0.0^{+0.0}_{-0.0}$ | $0.0^{+0.0}_{-0.0}$ | $0.0 \pm 0.0$ | 202.0 | $67.7^{+4.0}_{-3.7}$ | $0.0^{+0.0}_{-0.0}$ | $0.0^{+0.0}_{-0.0}$ | $0.0 \pm 0.0$ | 148.7 |
| Pokec | GRAIN | $58.3^{+5.9}_{-5.9}$ | $50.7^{+7.9}_{-7.8}$ | $55.8^{+8.3}_{-7.9}$ | $15.0 \pm 0.8$ | 0.1 | $97.2^{+1.6}_{-1.9}$ | $93.5^{+3.4}_{-4.2}$ | $96.3^{+1.9}_{-2.3}$ | $79.0 \pm 1.8$ | 0.2 |
| | DLG | $44.6^{+6.8}_{-6.2}$ | $11.3^{+16.0}_{-11.3}$ | $13.7^{+20.1}_{-13.7}$ | $0.0 \pm 0.0$ | 37.8 | $44.7^{+2.3}_{-2.3}$ | $2.2^{+3.1}_{-2.2}$ | $0.0^{+0.0}_{-0.0}$ | $0.0 \pm 0.0$ | 26.3 |
| | DLG $+A$ | $48.7^{+12.7}_{-8.6}$ | $39.1^{+18.7}_{-16.9}$ | $51.8^{+15.9}_{-17.8}$ | $1.0 \pm 0.2$ | 38.5 | $57.4^{+3.7}_{-3.9}$ | $69.5^{+3.6}_{-4.0}$ | $88.6^{+2.0}_{-2.1}$ | $0.0 \pm 0.0$ | 21.6 |
| | TabLeak | $49.6^{+8.7}_{-6.6}$ | $8.2^{+12.0}_{-8.2}$ | $5.6^{+9.3}_{-5.6}$ | $0.0 \pm 0.0$ | 177.5 | $50.8^{+12.4}_{-8.9}$ | $13.9^{+13.5}_{-12.3}$ | $7.9^{+11.9}_{-7.9}$ | $0.0 \pm 0.0$ | 204.5 |
| | TabLeak $+A$ | $49.9^{+4.1}_{-4.3}$ | $38.1^{+5.8}_{-5.9}$ | $58.9^{+6.1}_{-6.6}$ | $0.0 \pm 0.0$ | 216.0 | $52.6^{+3.3}_{-3.3}$ | $68.1^{+4.1}_{-3.9}$ | $82.7^{+4.0}_{-4.9}$ | $0.0 \pm 0.0$ | 254.5 |

**Effect of optimizer on baseline results** In our main experiments we presented results for the CiteSeer and Pokec datasets with the baselines running an SGD optimizer. In Tab. 9, we present results with the more stable LBFGS optimizer averaged across 10 reconstructions due to time limits. We see that the baselines show better performance, however, GRAIN still outperforms them and is less resource-consuming, requiring up to $100\times$ less runtime.

**Effect of model parameters on reconstruction quality** First, in Tab. 10 and Tab. 11 we demonstrate the performance of GRAIN under modifying the model parameters. We observe that neither changing in the number of layers nor the hidden dimension size of the GCN substantially affects the performance of GRAIN, while reaffirming the significant improvement over the baselines, even when they are given the graph connections as prior knowledge. We note that we only utilise the first 2 GCN layers even when $L > 2$, showing the robustness of our method.

Additionally, we note that GRAIN is not significantly impacted by the embedding dimension $d'$, as long as $n < d'$, consequently achieving similar scores, particularly for small graphs. We show the exact results in Tab. 11.

**Effect of span check threshold on filtering capabilities** We now investigate the effect of the choice for the $\tau$ threshold, used for filtering inputs using the span check method. We measure the ratio between the number of nodes and 1-hop building blocks that pass the filter, and the actual number of these blocks. We explore different values of $\tau$ in the range $[10^{-6}, 1]$, and evaluate this metric on 10 randomly chosen samples from the Tox21 dataset. We show in Fig. 6 that any $\tau \in [10^{-4}, 10^{-2}]$ results in essentially the same filtering results, and that thresholds in this interval perfectly recover the correct 1-hop building blocks.

**Adjacency matrix low-rankness effect on reconstructability** Further, in Fig. 3b we looked into how the rank-deficiency of the adjacency matrix $A$ affects how much of the input GRAIN might be able to recover. For different sizes of $A$, we measure what the Monte-Carlo probability of $A$ being full-rank, and the fraction of nodes we can recover, as computed per Thm. 5.1. This was done for synthetic graphs, where we sampled 100,000 symmetric binary matrices with varying probability of every 2 nodes being connected, as well as for all molecular graphs in the chemical datasets Clintox, Tox21 and BBBP. We show that Thm. 5.1 is crucial for understanding why GRAIN is effective, despite the probability of $A$ being full-rank being low. In particular, we highlight in Fig. 3b that GRAIN can recover an increasing fraction of nodes as $A$ grows.

Table 10: Results (in %) of GRAIN and the baselines in cases of different model parameters. Here $L$ is the number of GCN layers and $d'$ is the model's width. $L = 2, d' = 300$ is the original setting.

| | | GSM-0 | GSM-1 | GSM-2 | FULL |
|---|---|---|---|---|---|
| $L = 2$, $d' = 300$ (default) | GRAIN | $86.9^{+4.2}_{-5.7}$ | $83.9^{+5.2}_{-6.9}$ | $82.6^{+5.7}_{-7.4}$ | $68.0 \pm 1.7$ |
| | DLG | $31.8^{+4.5}_{-4.3}$ | $20.3^{+5.5}_{-4.8}$ | $22.8^{+6.6}_{-5.6}$ | $1.0 \pm 0.2$ |
| | DLG $+A$ | $54.7^{+3.9}_{-4.2}$ | $60.1^{+4.6}_{-5.2}$ | $76.7^{+3.6}_{-4.8}$ | $1.0 \pm 0.2$ |
| | TabLeak | $25.1^{+5.1}_{-4.3}$ | $12.4^{+5.5}_{-4.3}$ | $10.8^{+5.6}_{-3.9}$ | $1.0 \pm 0.2$ |
| | TabLeak $+A$ | $55.6^{+3.9}_{-3.9}$ | $57.7^{+4.1}_{-4.6}$ | $73.8^{+2.8}_{-3.5}$ | $1.0 \pm 0.2$ |
| $L = 3$, $d' = 300$ | GRAIN | $82.5^{+5.7}_{-7.7}$ | $80.7^{+6.3}_{-7.7}$ | $80.4^{+6.2}_{-7.8}$ | $63.0 \pm 1.6$ |
| | DLG | $20.3^{+4.3}_{-3.4}$ | $7.8^{+5.1}_{-3.3}$ | $8.2^{+5.3}_{-3.4}$ | $1.0 \pm 0.2$ |
| | DLG $+A$ | $43.0^{+3.7}_{-3.6}$ | $48.0^{+4.3}_{-4.5}$ | $66.0^{+3.7}_{-4.6}$ | $1.0 \pm 0.2$ |
| | TabLeak | $16.5^{+3.8}_{-2.9}$ | $8.8^{+4.4}_{-3.1}$ | $8.0^{+4.3}_{-3.0}$ | $1.0 \pm 0.2$ |
| | TabLeak $+A$ | $47.5^{+4.0}_{-4.2}$ | $48.1^{+4.8}_{-5.0}$ | $62.9^{+4.3}_{-4.4}$ | $1.0 \pm 0.2$ |
| $L = 4$, $d' = 300$ | GRAIN | $83.9^{+5.5}_{-7.4}$ | $82.8^{+5.9}_{-7.7}$ | $82.8^{+6.0}_{-7.9}$ | $64.0 \pm 1.6$ |
| | DLG | $14.1^{+3.8}_{-2.8}$ | $4.0^{+4.7}_{-2.2}$ | $4.8^{+4.9}_{-2.6}$ | $1.0 \pm 0.2$ |
| | DLG $+A$ | $39.1^{+3.7}_{-3.8}$ | $37.0^{+5.3}_{-5.4}$ | $55.6^{+5.0}_{-5.7}$ | $1.0 \pm 0.2$ |
| | TabLeak | $12.0^{+3.4}_{-1.9}$ | $2.1^{+4.3}_{-1.4}$ | $3.4^{+4.0}_{-1.7}$ | $1.0 \pm 0.2$ |
| | TabLeak $+A$ | $30.0^{+4.7}_{-4.0}$ | $27.3^{+5.9}_{-5.1}$ | $51.1^{+4.9}_{-5.3}$ | $1.0 \pm 0.2$ |
| $L = 2$, $d' = 200$ | GRAIN | $84.6^{+4.6}_{-6.4}$ | $81.4^{+5.8}_{-6.9}$ | $80.5^{+5.9}_{-7.2}$ | $62.0 \pm 1.6$ |
| | DLG | $30.8^{+4.5}_{-4.1}$ | $18.9^{+5.8}_{-4.9}$ | $22.2^{+6.7}_{-5.4}$ | $1.0 \pm 0.2$ |
| | DLG $+A$ | $50.3^{+4.2}_{-4.2}$ | $53.4^{+5.3}_{-5.9}$ | $68.7^{+4.9}_{-6.1}$ | $3.0 \pm 0.4$ |
| | TabLeak | $22.1^{+4.8}_{-3.7}$ | $10.3^{+5.3}_{-3.6}$ | $8.9^{+5.5}_{-3.6}$ | $1.0 \pm 0.2$ |
| | TabLeak $+A$ | $55.0^{+4.8}_{-5.0}$ | $62.1^{+4.9}_{-5.9}$ | $76.7^{+3.6}_{-4.7}$ | $1.0 \pm 0.2$ |
| $L = 2$, $d' = 400$ | GRAIN | $85.2^{+4.6}_{-6.1}$ | $81.5^{+5.4}_{-7.1}$ | $80.1^{+6.1}_{-7.5}$ | $63.0 \pm 1.6$ |
| | DLG | $35.1^{+4.9}_{-4.7}$ | $26.1^{+6.4}_{-5.6}$ | $25.0^{+6.9}_{-6.0}$ | $1.0 \pm 0.2$ |
| | DLG $+A$ | $57.6^{+3.9}_{-4.3}$ | $61.7^{+4.7}_{-5.5}$ | $72.5^{+4.3}_{-5.5}$ | $2.0 \pm 0.3$ |
| | TabLeak | $28.5^{+4.5}_{-4.0}$ | $17.1^{+5.4}_{-4.4}$ | $12.9^{+5.4}_{-4.0}$ | $1.0 \pm 0.2$ |
| | TabLeak $+A$ | $61.7^{+3.6}_{-3.7}$ | $62.6^{+3.6}_{-4.4}$ | $76.3^{+2.9}_{-3.3}$ | $1.0 \pm 0.2$ |

Table 11: Results (in %) of GRAIN with different embedding dimensions across a range of graph sizes

| | $n \leq 15$ | | | $16 \leq n \leq 25$ | | | $26 \leq n$ | | |
|---|---|---|---|---|---|---|---|---|---|
| | GSM-0 | GSM-2 | FULL | GSM-0 | GSM-2 | FULL | GSM-0 | GSM-2 | FULL |
| $d = 300$ | $93.0^{+3.4}_{-5.4}$ | $91.6^{+3.8}_{-6.3}$ | $81.9 \pm 1.7$ | $81.7^{+3.9}_{-4.8}$ | $74.8^{+5.8}_{-6.3}$ | $43.6 \pm 1.1$ | $50.1^{+6.8}_{-7.1}$ | $39.2^{+8.5}_{-7.7}$ | $5.1 \pm 0.6$ |
| $d = 128$ | $92.1^{+3.2}_{-5.0}$ | $92.3^{+3.9}_{-5.7}$ | $79.3 \pm 1.6$ | $81.4^{+5.2}_{-4.8}$ | $75.1^{+6.6}_{-6.6}$ | $43.6 \pm 1.1$ | $49.3^{+7.2}_{-6.5}$ | $38.8^{+8.7}_{-7.6}$ | $5.1 \pm 0.6$ |
| $d = 64$ | $92.2^{+3.0}_{-5.5}$ | $92.0^{+4.0}_{-5.9}$ | $79.3 \pm 1.6$ | $81.3^{+4.1}_{-4.7}$ | $75.5^{+5.8}_{-6.5}$ | $43.6 \pm 1.1$ | $48.6^{+7.4}_{-6.5}$ | $37.9^{+9.0}_{-7.7}$ | $5.1 \pm 0.6$ |
| $d = 32$ | $92.2^{+3.0}_{-5.5}$ | $91.7^{+3.6}_{-6.5}$ | $79.3 \pm 1.6$ | $81.7^{+4.0}_{-4.4}$ | $73.8 \pm 6.1$ | $43.6 \pm 1.1$ | $15.3^{+2.8}_{-4.4}$ | $13.3^{+2.5}_{-3.9}$ | $0.0 \pm 0.0$ |

## D  LIMITATIONS

GRAIN is the first algorithm to advance the field of gradient inversion for graph data, and we see significant potential for further development. However, our attack method currently assumes that the FL protocol includes node degree as a node feature. While this assumption holds in many GNN settings, relaxing it leads to reduced performance. Improving performance under relaxed assumptions is an important direction for future research.

Another key area for improvement is reducing the computational complexity of GRAIN, enabling it to scale to larger graphs and graphs with nodes of higher degree. We believe this to be a promising future direction, as there are many potential optimizations that could enhance the algorithm's

Table 12: Results (in %) of GRAIN tested on the Pokec social network dataset  (Rossi & Ahmed, 2015). 20 subgraphs were sampled for each of the size ranges 25-30, 30-40, 40-50 and 50-60

| $n$ | GSM-0 | GSM-1 | GSM-2 | FULL | Runtime[h] |
|---|---|---|---|---|---|
| 25-30 | $98.3^{+0.2}_{-0.4}$ | $95.1^{+0.5}_{-1.1}$ | $96.8^{+0.4}_{-0.9}$ | 17/20 | 0.17 |
| 30-40 | $83.1^{+2.3}_{-3.4}$ | $61.6^{+3.1}_{-3.0}$ | $79.4^{+2.7}_{-3.6}$ | 5/20 | 0.46 |
| 40-50 | $69.3^{+3.2}_{-3.8}$ | $38.0^{+4.7}_{-4.3}$ | $59.2^{+3.7}_{-4.0}$ | 2/20 | 0.64 |
| 50-60 | $32.7^{+4.8}_{-3.9}$ | $23.3^{+4.2}_{-3.5}$ | $41.2^{+4.6}_{-4.1}$ | 3/20 | 0.43 |
| Total | $70.9^{+6.2}_{-6.5}$ | $55.6 \pm 7.2$ | $69.2^{+6.4}_{-6.6}$ | 27/80 | 1.70 |

efficiency. Particularly promising is using data priors that have the potential to severely reduce the number of span checks that need to be performed by efficiently filtering impossible subgraphs.

Additionally, GRAIN relies on the assumption that the GNN architecture satisfies Asm. A.1. While this assumption holds for widely used GNNs like GCN and GAT, adapting our algorithm to support other GNN types is left for future work.

Moreover, as discussed in Sec. 5, GRAIN requires that $n < d'$ to maintain the low-rank nature of gradient updates. While this is a limitation, we believe it applies to many real-world scenarios, meaning that practical FL settings remain vulnerable to privacy risks.

Another assumption GRAIN depends on is that the weighted adjacency matrix $A$ is high-rank, with full reconstruction possible only if $A$ is full-rank. While this holds for GAT architectures, it is not always true for GCNs as presented in Fig. 3b. Relaxing the full-rankness assumption would be a crucial step toward better understanding privacy risks in FL for GNNs.

Figure 6: Ablation study on the span check filtering threshold $\tau$.

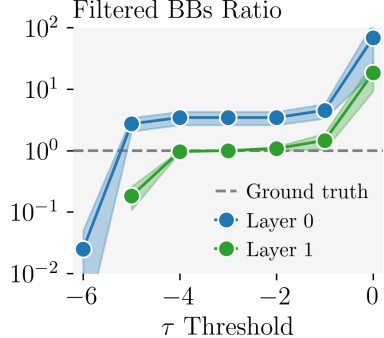

Finally, we leave the investigation of potential defenses against GRAIN as well as more complex federated learning protocols such as Federated Averaging for future work.

