# OpenReview forum: "GRAIN: Exact Graph Reconstruction from Gradients"
_ICLR.cc/2025/Conference — ICLR 2025 Poster_

### Official Review · Reviewer_t4xV · 2024-10-16

**Soundness:** 2
**Presentation:** 1
**Contribution:** 2
**Rating:** 5
**Confidence:** 3

**Summary:**

This paper proposes a method for gradient inversion attack on graph neural networks. This problem differs from previous gradient inversion attacks in that not only the node features but also the graph structures. The proposed method involves a recursive construction process that continually glues small building blocks to build large graphs. The authors propose some new metrics for this new task of gradient inversion attack on graphs, and show that the proposed method achieves state-of-the-art performance compared to other existing gradient inversion attacks.

**Strengths:**

1. This paper tackles a new problem of gradient inversion attack on graphs. More importantly, gradient inversion attack on graphs is different from GIA for other types of data, in that graphs not only have feature vectors, but also have graph structures.
2. This paper proposes a new evaluation metric to evaluate GIA on graphs, which would help future efforts in this field. At present, this research field is blank without any existing evaluation protocols, so the first such protocol or metric is nice to have.

**Weaknesses:**

1. The presentation of this paper requires significant improvements. The technical methodology of this paper is very hard to understand.

- The abstract contains a redundant sentence 'Federated learning allows multiple parties to train collaborative while'.

- It is suggested that the authors should clearly state the attack setting. For example, who is the attacker (server or client), what does the attacker know (model parameters, gradients, anything else).

- The authors should revise the use of the term 'degree'. It is used simultaneously to describe 'the number of edges connected to a node', as well as 'the number of hops of a subgraph'. This makes the paper very confusing to read. The second usage of degree can be replaced to something like 'hop'.

- For Theorem 3.1, the authors should define the concept of 'rowspan' and 'colspan', and briefly state the implication of the theorem. The same holds for Lemma 5.1.

- It is not clear how to obtain $T_0$.

2. This paper makes unrealistic assumptions about the problem and the data.

- In Lemma 5.1, the authors assume that $\tilde{A}$ is full rank. However, this is an unrealistic assumption, in that in practice, the adjacency matrix of most graphs are low-rank [1,2]. Therefore, for this Lemma to work, the authors either need to show that real-world graphs are almost full rank, or discuss what happens when the adjacency matrix is low-rank.

- In Page 4, Lines 183-190, the authors say that 'we leverage that the degree of a node is a widely used node feature', and subsequently assume that the degree is a known knowledge. However, this is not always true. When the node features are bag-of-word vectors, word embeddings, etc., the assumption may not hold. It is suggested that the authors should at least explicitly state the assumption.

- In Section 6.2, the authors say that 'we provide the attack with the correct number of nodes'. However, in practice, this is often not the case --- how can the attacker know the number of nodes before attacking? The authors should either justify the assumption, or discuss what happens when the attacker does not know the information.

- In Section 5.1 and Figure 2, the authors show that the gluing operation actually merges the same node $v$ in two building blocks, and merge edges similarly. However, I did not quite understand how the information of 'two nodes in two different building blocks correspond to the same node' is obtained. The authors should state how to obtain this information, as in practice, all reconstructed nodes are not given node indices and are only identified with features (which are inaccurate by themselves).

3. The authors' claim that the proposed method is 'exact' should be an overclaim. In fact, the proposed GRAIN can reconstruct 30-70% of all molecules (Table 1), which, in my opinion, is not sufficient to claim 'exact'. The authors should revise the claim to better fit the actual effectiveness of GRAIN.

[1] Graph Structure Learning for Robust Graph Neural Networks. KDD 2020

[2] Learning social infectivity in sparse low-rank networks using multi-dimensional hawkes processes. AAAI 2013.

**Questions:**

1. Please briefly explain the unclear parts stated in Weakness 1.

2. Please briefly discuss how the assumptions in Weakness 2 hold in practice, and what will happen when they do not hold.

3. In practice, graph data are often organized in a batch, and the gradients are an average of all samples in the batch. Does GRAIN assume batch_size=1? Will this change the effectiveness of GRAIN?

4. In the case with more complex/adaptive node-wise relations, such as Graph Attention Networks, will GRAIN still work? From my understanding of Lemma 5.1, $\tilde{A}$ plays an important role, and in GAT, the actual message passing topology is not $\tilde{A}$ .

**Details Of Ethics Concerns:**

Not needed.

---

> ### Author Response · Authors · 2024-11-24
>
> We thank Reviewer $\RFI$ for their constructive feedback. We are pleased that they recognize our paper addresses the novel problem of gradient inversion attacks on graphs and appreciate the introduction of the proposed evaluation metric for gradient inversion attacks on GNNs. Below, we address their questions and concerns in detail:
>
> **Q5.1: The presentation of this paper requires significant improvements.**
>
> We appreciate the reviewer’s feedback. We will address all writing concerns raised in the next revision. In the meantime, we provide a detailed threat model in Q.3 of the main response and address additional writing concerns below.
>
> **Q5.2: Can the authors define the concept of 'rowspan' and 'colspan'?**
>
> The rowspan of a matrix with row vectors $v_1,v_2,\dots, v_n$ refers to the set of vectors consisting of vector that can be constructed as a linear combination $\alpha_1 v_1 + \dots + \alpha_n v_n$. The definition of colspan is similar but for the column vectors of the matrix. We omit these definitions, as we believe the terms to be standard in linear algebra and, thus, we expect that most readers will be readily familiar with them.
>
> **Q5.3: Can the authors briefly state the implication of Theorem 3.1 and Lemma 5.1?**
>
> Theorem 3.1, originally introduced in Petrov et al. [1], demonstrates that when the number of true input vectors to a linear layer $Z=XW$ is less than its hidden dimension, one can efficiently verify whether a chosen input vector is among the true inputs with high probability by measuring its proximity to the subspace spanned by the columns of the weight gradient $\grad{W}$. Assuming discrete features, this allows us to create an efficient filtering procedure, by enumerating all possible inputs to a layer and measuring the proximity to the subspace of each.
>
> Lemma 5.1 generalizes Theorem 3.1 for layers of the form $Z=AXW$ to apply it to GCNs. It states that an input row of $X$ will lie in the column span of $\grad{W}$ if and only if the corresponding column of the adjacency matrix $A$ is linearly independent of the other columns. We discuss how this impacts the reconstruction capabilities of GRAIN in more detail in **Q.2**, and we will incorporate these clarifications in the next revision of the paper.
>
> **Q5.4: Can the authors clarify how GRAIN obtains the set $T_0$?**
>
> The set $T_0$ is constructed by considering all possible feature combinations. Specifically, since we assume each feature is discrete, these combinations can be enumerated by exploring all options in the cross product of each feature set. This process can be carried out by the adversary, as this information is part of the threat model, as explained in **Q.3** of the main response.Whenever this set is intractable to exhaustively compute, we instead recover the node features one-by-one by iteratively filtering the feature combinations. Please refer to our response to **Q1.1** of Reviewer $\RO$ for an in-depth explanation of the procedure.
> Q5.5 Can GRAIN handle real-world graphs, whose adjacency matrices are often low-rank?
>
> Thank you for the insightful question! We provide a thorough discussion of this topic in **Q.2** of the main response, where we relax the requirement for $A$ to be full rank. Additionally, as the size of the graph increases, we are able to recover more nodes on average. This improvement follows from the relaxation of Lemma 5.1, which has been adapted to better align with the requirements of GRAIN.
>
> **Q5.6 Could the authors include the assumption that GRAIN requires the degree as part of the feature vector, and show what happens without using this information?**
>
> Thank you for the suggestion! We have included the assumption when discussing the threat model in **Q.3**, which we plan to add to the next revision of the paper. That said, we have shown in **Q.6** that it is not an explicit requirement, but a way to make the task less computationally intensive, and show good initial results without it.

---

> ### Author Response · Authors · 2024-11-24
>
> **Q5.7 What do the authors mean by “providing the attack with the correct number of nodes” in Section 6.2?**
>
> We would like to note that this assumption is utilised *only for the baseline attacks*, as described in Section 6.2 of the paper. We do not claim it is a reasonable assumption, as the attacker has no easy way to recover this information. However, it is required to make the baselines applicable in this setting, as to setup the optimization variables $A$ and $\mathbf{X}$ the number of graph nodes must be known. As GRAIN does not make that assumption, this further highlights the effectiveness and practicality of our attack.
>
> **Q5.8 How does the gluing operation determine which nodes from different subgraphs are the same?**
>
> To perform the gluing operation, we emphasize that we are working with colored graphs, where each node is associated with a set of  features. This allows us to determine if two nodes might be the same by comparing their features and checking if their neighbors are compatible (i.e., the set of neighbors of one node is a subset of the neighbors of the other). We would like to highlight that because the recovered features are discrete, we are able to assert whether two feature vectors are equal with no margin of error. The gluing operation does not result in a correct matching in only 2 rare cases. One, during the construction of degree-$l$ blocks, we filter out the incorrect blocks using the span check. Two, during the DFS reconstruction of the entire graph, we assert that the relevant branch in the tree search will eventually be discarded.
>
>
> **Q5.9 Does the term “exact” accurately describe the capabilities of GRAIN?**
>
> We appreciate the reviewer’s question. We want to emphasize that Lemma 5.1 in our paper provides a theoretical guarantee for exact reconstruction of the input features of GCNs with high probability. To this end, we believe the term exact is justified. We acknowledge, however, that due to computational constraints on the part of the attacker full recovery of the underlying graphs may not be possible in all settings. Therefore, we will rephrase this as "eventually exact" to more accurately reflect the computational considerations.
>
> **Q5.10 Can GRAIN handle batches of size $B>1$?**
>
> As GRAIN is the first gradient inversion attack specifically targeting GNNs, it was initially designed for $B=1$. Consequently, recovering a batched input is more challenging but still achievable. We note that all building blocks can be recovered in the same way, as $B>1$ can be treated as a disconnected graph with $B$ components. During the tree search phase, each of the $B$ graphs would correspond to a leaf in the search space. We can then compute the gradient distance for all possible combinations of these leaves and return the one with a distance of 0. This problem can be simplified using the heuristics outlined in **Q.7**. In conclusion, we believe this procedure can be extended to handle $B>1$, but its implementation is left for future research.
>
> [1] Petrov, Ivo, et al. "DAGER: Exact Gradient Inversion for Large Language Models." arXiv preprint arXiv:2405.15586 (2024).

---

> ### Comment · Reviewer_t4xV · 2024-11-25
>
> I would like to thank the authors for their very detailed rebuttal. I would like to raise my rating to 5.

---

### Official Review · Reviewer_5Wij · 2024-10-26

**Soundness:** 3
**Presentation:** 3
**Contribution:** 3
**Rating:** 6
**Confidence:** 3

**Summary:**

This paper presents GRAIN, a novel gradient inversion attack designed for Graph Convolutional Networks (GCNs) in the federated learning (FL) setting. GRAIN leverages an efficient filtering mechanism to identify subgraphs of the input graphs, which are then pieced together using a depth-first traversal to recover the full original graph. This method enables the exact reconstruction of graph structures and node features from gradient updates clients share with the federated server. The main contribution is extending gradient inversion attacks to GCNs under harder FL settings, which is a significant step toward understanding the privacy vulnerabilities of federated learning when applied to graph-structured data. The introduction of new evaluation metrics for graph similarity (e.g., GRAPH-N) is also an excellent contribution, providing valuable insights into partial and exact graph reconstructions. Lastly, this paper presents extensive rigorous experiments on molecular data, demonstrating that GRAIN significantly outperforms existing baseline attacks regarding exact graph reconstruction accuracy in the chemical domain. The empirical performance convincingly highlights client privacy risks associated with FL for graph-structured data, making this work relevant to a broad audience.

Overall, I am leaning toward acceptance of this paper. The novelty of GRAIN, its strong experimental results, and its contribution to the discussion on privacy risks in federated learning make it a valuable and impactful addition to the conference. However, there are some limitations and concerns that require further clarification and improvement (see weaknesses).

**Strengths:**

1. Technical novelty.
2. Strong experimental results.
3. Contribution to the discussion on privacy risks in federated learning.

**Weaknesses:**

One major limitation is that GRAIN struggles with larger graphs due to the high computational cost. As mentioned in the paper, the method times out for large molecules. Have you considered any strategies for reducing the computational complexity of GRAIN, such as parallelizing the depth-first search or using pruning techniques? Furthermore, more detailed comparisons of the convergence time of GRAIN versus the baseline methods might be beneficial, particularly for larger graphs. Could you provide more quantitative details on the computational overhead of GRAIN in comparison to other baselines?

I noticed several similarities between the algorithms and figures in this paper and those in your reference DAGER. I would appreciate more clarification on how this paper differentiates itself and introduces novel contributions beyond the existing work. Highlighting the differences, extensions, and innovations more clearly and explaining your considerations on adaptions might strengthen the paper's contribution.

Finally, in the experimental section, the paper notes that GRAIN cannot recover multivalent interactions, an edge property that GCNs not able to capture. However, given multivalent interactions are associated with features like "valence structures" in the MoleculeNet benchmark datasets, whether other node features such as valence structures be fully used in graph filter and node-feature recovery? This raises the question of whether GRAIN has limitations in effectively utilizing feature vectors of the input nodes. It would be helpful if the authors could clarify whether GRAIN could recover some other complex critical chemical characteristics with all feature vectors and the reasons behind them.

For the experiments, the following should be addressed.

1.	It would be helpful to see more experiments that vary the value of chosen threshold tau in the span check mechanism to better understand its role in filtering performance.

2.	For the scenario where exact reconstruction is not achieved, what proportion of nodes and edges are typically misplaced, and how? Additional discussion on this and how it affects real-world privacy concerns would provide a more comprehensive picture of the method's practical implications.

3.	As the proposed graph similarity metric is novel and potentially valuable, I have some concerns regarding the fairness and comparability of the results. Specifically, how do you ensure that the comparison is fair, given that the baseline methods were not specifically designed for graph-structured data? Additionally, it would be helpful to include a more detailed explanation of why common, widely used metrics were not used for comparison in the meantime. This clarification would strengthen the validity of your results and provide a clearer understanding of how the proposed metric aligns with established evaluation standards.

Minor comments:
1)	In the Abstract, there is redundancy in the first sentence (e.g. “Federated learning allows multiple parties to train collaboratively while only Federated learning…”).
2)	In the third line in 5.1, the symbol is a subset instead of an overlap.
3)	In 5.1, the third sentence in Additional structure-based filtering and likelihood ordering, the grammar of “Specifically, we for every…” is wrong. The last sentence in the same subsection “lines 3–13 of Algorithm 1” should be 3-14.

**Questions:**

See weaknesses.

---

> ### Author Response · Authors · 2024-11-24
>
> $\newcommand{\RF}{\textcolor{purple}{5Wij}}$We thank reviewer $\RF$ for the positive review. We are happy to read that the reviewer acknowledges the technical novelty of our paper, the strong experimental results and the overall contribution to understanding the privacy risks of GNNs to federated learning. We offer additional clarification on their questions below:
>
> **Q4.1: Can GRAIN scale to larger graphs?**
>
> Thank you for the question! It is important to note that GRAIN will eventually find the correct client through an exhaustive tree search, which can become computationally expensive for larger graphs. To address this, we believe that using heuristics tailored to each data type would be most effective. For example, this could involve imposing a prior on the behavior of cycles in molecular data or specifying an order for connecting building blocks. We provide further details in **Q.7**, along with promising initial results, and we plan to include additional experiments in the next revision of the paper.
>
> **Q4.2: Can you provide a runtime comparison to prior work, especially w.r.t. graph size?**
>
> Yes! As shown in Table 9 in Appendix B.2, GRAIN achieves significantly better results while running for a comparable amount of time to the Tableak attack (14-24 hours versus 12-15 hours). For the baselines, the reported times reflect the duration each iteration ran until convergence, ensuring a fair evaluation.
>
> **Q4.3: Can you clarify the paper’s novel contributions over existing work?  Highlight the differences, extensions, and innovations and explain your considerations on adaptions.**
>
> The paper’s novel contributions over existing work, particularly over Petrov et al. [1], include the introduction of GRAIN as the first gradient inversion attack on GNNs, showing GNNs are vulnerable to gradient leakage attacks. Key innovations include overcoming the challenges posed by the unknown adjacency matrix $A$ through local subgraph reconstruction, the extension of Petrov et al.'s theory to GNNs, and an efficient GPU implementation for an efficient space exploration. Additionally, GRAIN improves the recovery of input features even when $A$ is rank-deficient, as we have determined from Lemma 5.1. These advancements make GRAIN more effective and accurate than previous methods.
>
>
> **Q4.4 Can GRAIN be adapted to better utilise chemical prior information on the recovered graph node features, to for example extract edge features such as valence from the per-node valence structures features?**
>
> Thank you for the insightful question! In our implementation we followed Rong et al. [2], including the following features: atom type, formal charge, number of bonds, chirality, number of bonded hydrogen atoms, atomic mass and aromaticity, meaning we utilise a smaller feature set compared to MoleculeNet. If "valence structures" were part of the input, we believe GRAIN could recover bond characteristics by comparing discrepancies between the number of connections and the valence.
>
> Further, we have observed that certain properties within our feature set, such as bond types (inferred from hybridization) or the location of aromatic rings (via the aromaticity feature), can indeed be recovered and used. In fact, these features could even help speed up the algorithm, as outlined in **Q.7** of the main response. However, we deliberately avoided using priors on the data to maintain a more generalizable framework.
>
> While we assert that GRAIN can recover the vast majority of complex molecule properties an attacker might be interested in, based solely on the recovered adjacency matrix $A$ and input feature $\mathbf{X}$, determining what chemical information is theoretically recoverable for particular choice of input features, and what isn’t is an interesting avenue for future research.
>
> **Q4.5: Can you provide experiments where you vary $\tau$?**
>
> Certainly! We demonstrate that the $\tau$ parameter is robust, as the results remain consistent across a wide range of values. To illustrate this, we measure the ratio between the number of nodes and degree-1 building blocks that pass the filter at a given threshold, compared to the actual number of these blocks, using 10 samples from the Tox21 dataset. As shown in Figure 5 in Appendix B.2, any value of $\tau \in [10^{-4}, 10^{-2}]$ yields nearly the same number of recovered nodes and the correct number of degree-1 building blocks. This suggests that our choice of $\tau = 10^{-3}$, the midpoint of this range, is an optimal selection for the hyperparameter.

---

> ### Author Response · Authors · 2024-11-24
>
> **Q4.6: Can you provide a qualitative analysis of the reconstruction quality of partially reconstructed graphs?**
>
> We have observed that the reconstructed graphs primarily comprise of degree-2 neighbourhoods which are part of the original graph. This typically indicates that our reconstruction shares a large common subgraph with the client input. This finding is further supported by the results of the human evaluation described in **Q.4**. As shown in Table 7 in Appendix B.1, our partial reconstructions are considered more significant than the metric alone would suggest, signifying substantial information leakage. In contrast, high-scoring examples from the DLG attack were rated as essentially uninformative.
>
>
> **Q4.7: Why are the prior reconstruction quality metrics insufficient to measure the graph reconstruction quality? How did you ensure that the metrics you introduced are fair with respect to the baseline attacks?**
>
> Thank you for the excellent question! We discuss this in detail in **Q.4** of the main response. To summarize, we developed a new set of metrics specifically designed to compare colored graphs, ensuring that isomorphic graphs receive perfect scores. To validate the fairness of our approach, we both provided arguments on how we relax the problem for the baseline models and conducted a user case study comparing our metrics with the preferences of three experts. This study confirmed that our metrics are representative, as shown in Table 6 in Appendix B.1.
>
> [1] Petrov, Ivo, et al. "DAGER: Exact Gradient Inversion for Large Language Models." arXiv preprint arXiv:2405.15586 (2024).
>
> [2] Rong, Yu, et al. "Self-supervised graph transformer on large-scale molecular data." Advances in neural information processing systems 33 (2020): 12559-12571.

---

> ### Author Response · Authors · 2024-11-26
>
> We would like to express our gratitude to the reviewer for acknowledging the importance of our work, the significance of our results, and for helping us improve our paper. We believe we have responded extensively to all the concerns and questions raised by them, including providing the additional experimental results requested. With the end of the discussion period approaching, we kindly request that the reviewer informs us of any additional questions or unresolved points, so we can address them in the paper. Additionally, we ask them to confirm they have read our response.

---

> ### Author Response · Authors · 2024-11-29
>
> We thank reviewer $\RF$ once again for the valuable feedback, we kindly direct you to our detailed response in **Q.8** of the main rebuttal, where we conducted further experiments on scalability, showing GRAIN can reconstruct graphs up to $\leq 60$ nodes.

---

### Official Review · Reviewer_HRGw · 2024-10-30

**Soundness:** 2
**Presentation:** 3
**Contribution:** 2
**Rating:** 6
**Confidence:** 3

**Summary:**

The main contribution of this paper is that it is the first to address gradient inversion attacks on graph data, highlighting privacy risks in graph federated learning. Based on the theorem from Petrov et al. (2024), the authors propose a method capable of reconstructing both the graph structure and node features. They reconstruct the graph step-by-step, from low-degree to high-degree levels, with each step involving the filtering of unlikely candidates through a span check. Finally, by leveraging degree information in the features, they generate the final prediction using a depth-first search (DFS) algorithm to combine the remaining candidates.

In experiments, the proposed method outperforms other baselines. The authors also conduct a hyperparameter analysis by adjusting L and d', representing the number of GNN layers and the hidden dimension, respectively. Additionally, they test varying numbers of nodes, identifying limitations as discussed in Section 7.

**Strengths:**

1. As the authors mention, this paper addresses a crucial yet unexplored problem: gradient inversion attacks in graph federated learning.
2. Constructing both the graph structure and features is a challenging task, but it is interesting to see experimental results that accurately reconstruct the input graph, even if restricted to chemistry graphs.
3. This paper suggests future directions for gradient inversion attacks in the graph domain, as discussed in Section 7 (Limitations).
4. The paper is well-written and easy to follow.

**Weaknesses:**

1. I agree on the importance of exploring gradient inversion attacks in the graph domain, and this could be a notable contribution of the paper as it is the first to address this issue. However, the scope is quite limited in terms of graph types. Specifically, the paper focuses only on chemistry datasets. To fully substantiate its contribution as a pioneering work in gradient inversion attacks on graph data, the authors should demonstrate that this method is applicable to other types of graph datasets, such as citation networks (e.g., Cora, PubMed) or transaction networks (e.g., Elliptic). This is especially important since molecular graphs may be relatively less privacy-sensitive than other graph types (e.g., transaction networks), and federated learning in molecular graphs may be less common. I suggest that the authors provide additional evaluations on other categories of graph datasets.

2. As shown in Table 2, the proposed method only works well when the input graph has a limited number of nodes (i.e., $\leq 25$). I understand that, given the step-by-step approach of combining building blocks, this limitation leads to higher computational costs and a greater risk of error as the graph size increases.

3. The claim that using the degree as a feature is widely adopted in training GCNs is not entirely convincing. Using degree values as features is more common in graphs lacking attributes or in structural learning tasks. However, since this paper addresses gradient inversion attacks in federated learning, it should consider more practical settings where degree values may not be used as features. How might this method be adapted if degree information is not available as a feature?

**Questions:**

1. In Table 3, $d' = [200, 300, 400]$ represents quite a large hidden dimension for training graphs with a small number of nodes (i.e., $\leq 25$). I wonder if this method will still work when the hidden dimension is much smaller (e.g., 32, 64, 128).

2. As I mentioned in W1, I suggest that the authors provide additional evaluations on other categories of graph datasets.

---

> ### Comment · Reviewer_HRGw · 2024-11-22
>
> Thank you for the authors' rebuttal. Most of my concerns have been addressed, and I am inclined to raise my score.

---

> ### Author Response · Authors · 2024-11-24
>
> $\newcommand{\RTH}{\textcolor{green}{HRGw}}$We thank reviewer $\RTH$ for their positive feedback. We are excited to read that the reviewer acknowledges the importance of exploring the field of gradient inversion attacks in the graph domain and that our paper is a notable contribution as the first one addressing the issue. We are further glad to see the reviewer finds our experimental results interesting and accurate. Finally, we were glad to read that the reviewer found our main rebuttal to have addressed their main concerns with the paper. We address any outstanding questions below:
>
> **Q3.1: Is GRAIN applicable to different types of graph data?**
>
> Yes! We have successfully applied GRAIN to the Citeseer citation network, which contains nodes with over 3,700 binary features. This demonstrates that GRAIN scales effectively to large feature sets and diverse dataset types. Our results include scores exceeding 70% and a full reconstruction rate above 60%. For details on how these experiments were conducted, please refer to **Q.5** in the main response.
>
>
> **Q3.2: Can GRAIN scale to graphs with more than 25 nodes?**
>
> Yes! As we show in **Q.7**, we have sufficient information encoded in the gradients, such that after enough time, the exact graph will be recovered. However, an exhaustive tree search is computationally expensive. To this end. we believe that there are ways to alleviate the majority of the inefficiencies in our algorithm through dataset-specific heuristics, which would allow GRAIN to scale to larger graphs. For example, we can leverage certain chemical properties that would restrict certain branches during the building phase for molecular data. We have discussed a set of directions that we find promising in **Q.7** of the main response. We have shown promising initial results and plan to include experiments that show further improvements in the next revision of the paper.
>
> **Q3.3: How might this method be adapted if degree information is unavailable as a feature?**
>
> Yes! We believe that many of the inefficiencies in our algorithm can be mitigated through heuristics tailored to the data type, enabling GRAIN to scale to larger graphs. For instance, in molecular data, specific chemical properties can be leveraged to prune certain branches during the building phase. We have outlined promising directions in **Q.7** of the main response and plan to include experiments demonstrating such improvements in the next revision of the paper.
>
> **Q3.4: Can you extend Table 3 with experiments on GCNs with smaller hidden dimension?**
>
> Yes! Since GRAIN only requires the embedding dimension $d^\prime$ to be larger than the number of nodes $n$, the proposed dimensions of 32, 64, and 128 do not significantly affect the results. Specifically, there is no statistically significant difference in scores for $d^\prime = 64$ and $d^\prime = 128$, and only larger graphs (with $\geq 25$ nodes) are impacted when $d^\prime = 32$. A comprehensive set of metrics for all experiments can be found in Table 8 of Appendix B.2.

---

> > ### Comment · Reviewer_HRGw · 2024-11-25
> >
> > Thank you for providing additional details in response to my questions. The rebuttal highlights the potential for expanding the application of this work to other graph datasets. However, the high computational complexity remains a limitation of this approach. Despite this, I believe the work holds promise and has the potential to pave the way for further advancements in the field of gradient inversion attacks within the graph domain. I will maintain my score at 6.

---

> ### Author Response · Authors · 2024-11-29
>
> We thank reviewer $\RTH$ once again for the valuable feedback and for engaging with our rebuttal. We kindly direct you to our detailed response in **Q.8** of the main rebuttal, where we conducted further experiments on scalability, showing GRAIN can reconstruct graphs up to $\leq 60$ nodes.

---

### Official Review · Reviewer_d7Q7 · 2024-11-02

**Soundness:** 3
**Presentation:** 2
**Contribution:** 2
**Rating:** 5
**Confidence:** 3

**Summary:**

This paper presents GRAIN, the first exact reconstruction attack specifically designed to target GCNs. By leveraging the low-rank structure of GCN layer updates, GRAIN can accurately reconstruct both the graph structure and the associated node features from the gradients shared under federated learning setting. This attack demonstrates a significant privacy vulnerability in GCN training within the federated learning framework.

**Strengths:**

1. It is the first work that explores extracting the structure of a graph from gradients.
2. The framework demonstrates promising performance across different scenario.

**Weaknesses:**

1. The article seems to be largely based on the theories presented in [1] and then adapted it for GCNs, particularly Theorem 3.1. However, for readers unfamiliar with this work, it may be quite challenging to comprehend. We hope the authors can provide more discussion on why the filtering mechanism can be effective.
2. Following the previous question, what is the difficulties encountered when applying the methods from [1] to graph data? A detailed explanation of the challenges encountered during this adaptation, including any limitations or obstacles that were overcome, would help clarify the novelty of the work presented.
3. The article uses the degree of a node as a node feature; however, this approach may limit the application scope. For instance, in the case of chemical molecules mentioned in the article, relevant features are more likely to include chemical properties. We suggest that the authors discuss this point and conduct further experiments.

[1] DAGER: Exact Gradient Inversion for Large Language Models.

**Questions:**

see Weaknesses

---

> ### Author Response · Authors · 2024-11-24
>
> $\newcommand{\Rt}{\textcolor{blue}{d7Q7}}$We thank reviewer $\Rt$ for the positive feedback and are happy that they appreciate GRAIN’s scalability across different scenarios. We further address their questions below:
>
> **Q2.1: Can the authors provide a more detailed discussion on why the filtering mechanism from Petrov et al. can be effective in the context of this paper?**
>
>
> As explained in Petrov et al. [1], the low-rank subspace spanned by the rows of the input $\mathbf{X}$ has hypervolume 0, and therefore, a random vector in $\mathbb{R}^{d}$ almost surely does not lie in it. As both Petrov et al. and GRAIN deal with a large but countable number of possible embedding vectors for the input of each layer, which can be considered random, the filtering procedure simply checks if any of them are in the span. Those that are in the span are almost surely the correct inputs to the layer, as the probability of them being wrong is essentially 0. Please refer to the proof of Theorem 5.2 in Petrov et al. [1] for more details.
>
> **Q2.2: What are the difficulties encountered when applying the methods from Petrov et al. to graph data?**
>
> We refer the reviewer to **Q.1** in the main response, where we outline the graph-specific challenges GRAIN encounters and how it tackles them, including the simultaneous recovery of the input node features and adjacency matrix, as well as the adaptation of the span-check procedure from Petrov et al. to handle this dual recovery. For the later, in particular, compared to Petrov et al. we develop a new theoretical understanding for how to handle rank-deficient adjacency matrices $A$, which is the key to explaining GRAIN’s efficiency on real-word graphs (see **Q.2**). We will ensure these clarifications are incorporated into the paper.
>
> **Q2.3: Does GRAIN rely on node in-degree to be part of the feature vector, and does that limit the applicability of the attack?**
>
> Thank you for this question! For a broader discussion and supporting experiments on this topic, please see **Q.6** in the main response. In summary, while incorporating this feature makes our attack more computationally efficient, it is not a mandatory requirement for applying GRAIN.
>
> [1] Petrov, Ivo, et al. "DAGER: Exact Gradient Inversion for Large Language Models." arXiv preprint arXiv:2405.15586 (2024).

---

> ### Author Response · Authors · 2024-11-26
>
> We would like to express our gratitude to the reviewer for the crucial questions they posed in their review. We believe we have responded to them extensively in the main response, and have summarized the answers above. With the end of the paper revision period approaching, we kindly request that the reviewer informs us of any additional questions or unresolved points, so that we can incorporate them in the paper if needed. Additionally, we ask them to confirm they have read our response and to consider updating their review accordingly.

---

> > ### Comment · Reviewer_d7Q7 · 2024-11-28
> >
> > Thank you for the authors' response in providing more details. While my concern has been partially addressed, the issue regarding the clarification of the method, scenarios, and limiting the application scope, as mentioned by another reviewer, seems to be a problem with this article. I will maintain my score at 5.

---

### Official Review · Reviewer_61N5 · 2024-11-03

**Soundness:** 3
**Presentation:** 1
**Contribution:** 2
**Rating:** 6
**Confidence:** 4

**Summary:**

This paper proposes GRAIN, a gradient inversion attack on GNNs. GRAIN identifies subgraphs within gradients and filters them iteratively by layer to construct the full input graph through a depth-first search approach. New evaluation metrics are introduced to measure similarity between reconstructed and original graphs for graph gradient inversion attacks. GRAIN has a desirable performance on the molecule benchmarks.

**Strengths:**

- This paper proposes the first gradient inversion attack on GNNs. This topic is interesting and unexplored. This paper could inspire further investigation on this topic.
- A new metric is proposed to measure the similarity between graphs.
- The proposed GRAIN achieves a desirable performance compared with existing attacks.

**Weaknesses:**

- It is mentioned that $\mathcal{T}_0$ is the cross-product of all possible feature values. It seems to be impractical for general attributed graphs and only possible for molecular graphs where the node features are in a small set with low dimensionality.
- It would be helpful to include a Threat Model section to introduce the attack settings, such as the adversary's knowledge, capability, and objective. What is the function $f$ in Algorithm 2? Is it the exact target GNN model?
- The introduction mentions federated learning as a practical scenario of gradient inversion attacks. However, the methodology part does not include any specific FL settings (such as local models and global models). Is FL a necessary requirement for implementing GRAIN?
- The challenge of gradient inversion attacks on GNNs is understated. Why GRAIN is not a trivial adaptation of existing methods [1] to GNNs and what unique challenge does GRAIN overcome while other methods fail to?
- The introduction of proposed algorithm is not clear enough. It would be better to introduce the detailed algorithm following a single direction (e.g., from input to output). And it would be helpful to add a notation table.


[1] Petrov, Ivo, et al. "DAGER: Exact Gradient Inversion for Large Language Models." arXiv preprint arXiv:2405.15586 (2024).

**Questions:**

Please see Weaknesses.

---

> ### Author Response · Authors · 2024-11-24
>
> $\newcommand{\RO}{\textcolor{red}{61N5}}$$\newcommand{\Rt}{\textcolor{blue}{d7Q7}}$$\newcommand{\RTH}{\textcolor{green}{HRGw}}$$\newcommand{\RF}{\textcolor{purple}{5Wij}}$$\newcommand{\RFI}{\textcolor{orange}{t4xV}}$$\newcommand{\grad}[1]{{\tfrac{\partial\mathcal{L}}{\partial #1}}}$$\def\colspan{{\text{ColSpan}}}$We thank the reviewer $\RO$ for the constructive feedback and are glad to read that the reviewer finds our paper could inspire future investigation on the topic. We are pleased that the reviewer acknowledges GRAIN's strong performance compared to existing attacks. We address their concerns in greater detail below::
>
> **Q1.1:  Does the exhaustive search GRAIN performs over the node features to generate T^0 pose a scaling issue for it?**
>
> Thank you for this question! While $\mathcal{T}_0$ may indeed be large when explored exhaustively, we are also able to reconstruct it in a step-by-step manner by filtering each feature individually. If the possible values for each one-hot encoded feature belong to sets $ \mathcal{F}_1, \mathcal{F}_2, …, \mathcal{F}_f$, the process proceeds as follows:
>
> 1. For the first feature set $ \mathcal{F}_1 $ of size $f_1$, we filter the correct feature vectors $\mathcal{F}_1^*$ using the span-check on the row-wise truncated gradient $\grad{W}[:{f_1}]$. This is possible by applying Lemma 5.1, as $\mathbf{X}[{i, :f_1}] \in  \colspan(\grad{W}[{:f_1}])$.
>
> 2. We then apply the same filtering procedure iteratively for each subsequent feature set $\mathcal{F}_k$ by combining the feature set with the filtered vectors from the previous step.
> 3. By induction, this approach allows us to construct $\mathcal{T}_0^* = \mathcal{F}_f^*$.
>
> We applied this method in practice on the Citeseer dataset, which contains over 3,000 binary features. Without our approach, $\mathcal{T}_0$ would have a size exceeding $2^{3,000}$, making exhaustive exploration infeasible. The results of this application are detailed in our answer to **Q.5** in the main response.
>
> **Q1.2: What is GRAIN’s threat model w.r.t. adversary knowledge, capabilities and objectives?**
>
> GRAIN follows the standard honest-but-curious threat model, where the server as the adversary aims to recover the client input from the reported gradients, while adhering to the protocol. A more thorough is provided in **Q.3** in the main response.
>
> **Q1.3: What does the set of functions $f_l$ in Algorithm 2 denote?**
>
> $\\{f_l\\}_{l\in[1,L]}$ is the set of functions that map the input of the $l$-th layer to the output of the $l$-th layer of the model. Additional clarifications of this and related notations have been included in Table 5 of Appendix A.
>
>
> **Q1.4: Are GRAIN and gradient inversion attacks in general FL specific?**
>
> Yes, in principle, gradient inversion attacks such as GRAIN are applicable to gradients computed on any model. However, maliciously obtaining gradients from a source that has computed and shared them for privacy-preserving reasons is unlikely to occur outside of an FL setup. This, in turn, means that our threat model detailed in **Q.1** in the main response is not well motivated outside of the FL setting. In case we misunderstood the reviewer’s intended question, we ask them to clarify what they meant by FL being a necessary requirement for GRAIN.
>
> **Q1.5: The challenge of gradient inversion attacks on GNNs is understated. Can the authors clarify what unique challenges GRAIN overcomes compared to the prior work?**
>
> Thank you for the suggestion. As detailed in **Q.1**, gradient inversion on GNNs presents unique challenges, such as simultaneously recovering a discrete input $\mathbf{X}$ and an adjacency matrix $A$, as well as the need for an efficient method to achieve this. We will ensure the paper is updated to include this discussion from **Q.1**.
>
> **Q1.6: The structure of the technical description of GRAIN can be improved. Can you add a notation table to the paper?**
>
> Thank you for the suggestion. We will overhaul the structure of the technical presentation of the paper in the next revision. For now, we supply the notation table in Table 5 in Appendix A.

---

> ### Author Response · Authors · 2024-11-26
>
> We would like to express our gratitude to the reviewer for their suggestions for improving the presentation of our paper and for acknowledging the novelty of our work. We believe we have responded extensively to all the concerns and questions raised by them. With the end of the paper revision period approaching, we kindly request that the reviewer informs us of any additional questions or unresolved points, so that we can incorporate them in the paper if needed. Additionally, we ask them to confirm they have read our response and to consider updating their review accordingly.

---

> ### Comment · Reviewer_61N5 · 2024-11-27
>
> I thank the authors for providing a detailed response. I am glad to see that GRAIN works for a broader range of graph data especially for those with a large feature dimensionality. However, the increase of feature number can bring an exponential increase of computation, still posing challenges even for basic real-world scenarios. It would be helpful to also provide the running time of GRAIN on the citeseer dataset.
>
> Additionally, the provided threat model information can largely help readers to understand the background of gradient inversion attack. However, given that I am not an expert in gradient inversion attacks, I feel like the assumption of data structure knowledge for the server might be too strong. The central server only needs to aggregate the received gradients, which does not necessitate the access to the data.
>
> Based on the new experimental results and further clarification provided, I am willing to increase my score to 5.

---

> ### Author Response · Authors · 2024-11-29
>
> We sincerely thank the reviewer for engaging with our rebuttal and acknowledging our detailed responses. Below, we address their remaining concerns:
>
> **Q1.7 Can GRAIN handle the exponential increase in the size of the feature combinations set $\mathcal{T}\_0$ when the number of options per node feature grows, and how does this exponential increase affect the practical runtime?**
>
> We acknowledge that naively enumerating the entire set  $\mathcal{T}\_0$ of possible feature combinations is indeed exponential in the number of node features. In **Q.1** in the main rebuttal, we show a simple modification to the GRAIN algorithm, where instead of generating the full $\mathcal{T}\_0$ we recover each feature after the $n$-th one in a feature-by-feature manner, alleviating the issue in practice. However, if many ordinal features are used, each with many possible values, this can still cause GRAIN to be intractable in practice. We demonstrate experimentally below that by exploring ordinal features at the end of our feature-by-feature filtering procedure, however, we can help alleviate this issue and make the process tractable again. The intuition is that when these features are explored at the end, if there are at least $n$ other features (including one-hot-encoding) the ordinal features will be explored one after the other. Further, each of their values will have to only be combined with a small set of plausible already-filtered vectors.
>
> We illustrate this in a set of experiments in the same setting as the experiments conducted over the Citeseer dataset in **Q.5** in the main response. We augmented the dataset with five discrete features, each containing 3,000 options, assigning a random value for each node. These features were handled by ordering them such that those with the highest number of possible values were recovered last. Using the heuristic described in **Q.7**, we achieved a GRAPH-1 score of $75.1^{+5.5}\_{-5.7}$​, representing an improvement over the previously reported score of $69.1$. This increase stems from a greater reduction in false positives passing through the span check. We report a runtime of 3.4 hours for both the new experiment, and the original described in **Q.7** with the heuristic, compared to 10.6 hours without the heuristic, which also yielded worse results (as detailed in **Q.7**). These experiments were notably faster than those conducted on the chemical datasets due to two factors: the efficiency of our feature-by-feature recovery approach and the tree search algorithm’s improved handling of nodes with unique feature vectors. These results underscore the practicality of our method and highlight GRAIN’s robustness and potential.
>
> **Q1.8 Why is the assumption that the server has knowledge of the data structure justified?**
>
> Thank you for raising this important point! In Federated Learning, the most common setup involves a central server coordinating communication with all clients. For a client to participate in the protocol, it must adhere to a shared data structure to ensure proper training, by making sure that input features correspond to the same information across participants. This is typically achieved by the server informing clients about the data structure, which necessitates the server’s prior knowledge of it. Concealing this information from the server would require a decentralized communication mechanism among clients, which is rarely adopted in practice, and poses additional risks when clients are malicious.
>
>
> Furthermore, this assumption is widely used in other works in the gradient leakage field. One such example is one of our baseline models, Tableak [6], which requires knowledge of which features are discrete (and what values they can take), and which are continuous. Similarly, every attack on textual data requires that the attacker have full knowledge of the tokenizer [7, 8, 9, 10], rely on the attacker knowing the tokenizer to map recovered embeddings or tokens back to words.
> We will update our threat model to include this discussion.
>
> [6] Vero, Mark, et al. "TabLeak: Tabular data leakage in federated learning." arXiv preprint arXiv:2210.01785 (2022).
> [7] Deng, Jieren, et al. "Tag: Gradient attack on transformer-based language models." arXiv preprint arXiv:2103.06819 (2021).
> [8] Petrov, Ivo, et al. "DAGER: Exact Gradient Inversion for Large Language Models." arXiv preprint arXiv:2405.15586 (2024).
> [9] Balunovic, Mislav, et al. "Lamp: Extracting text from gradients with language model priors." Advances in Neural Information Processing Systems 35 (2022): 7641-7654.
> [10] Fowl, Liam, et al. "Decepticons: Corrupted transformers breach privacy in federated learning for language models." arXiv preprint arXiv:2201.12675 (2022).

---

> > ### Comment · Reviewer_61N5 · 2024-12-02
> >
> > I thank the authors for their further clarification. My primary concern regarding the scalability has been largely alleviated. Additionally, the explanation of threat model makes sense to me. I have raised my score.

---

### Author Response · Authors · 2024-11-22
**Main Response to ICLR 2025 Official Reviews**

$\newcommand{\RO}{\textcolor{red}{61N5}}$$\newcommand{\Rt}{\textcolor{blue}{d7Q7}}$$\newcommand{\RTH}{\textcolor{green}{HRGw}}$$\newcommand{\RF}{\textcolor{purple}{5Wij}}$$\newcommand{\RFI}{\textcolor{orange}{t4xV}}$$\newcommand{\grad}[1]{{\tfrac{\partial\mathcal{L}}{\partial #1}}}$$\def\colspan{{\text{ColSpan}}}$We thank the reviewers for their valuable input, as we strongly believe that it has made our paper stronger. We are delighted to read that reviewers find gradient inversion attacks on graph neural networks an important ($\RTH$), unexplored($\RO, \Rt, \RTH$), and interesting($\RO$) topic, and that our results constitute a significant step toward understanding the privacy vulnerabilities of federated learning when applied to graph-structured data ($\RF$) and could inspire further research in the area ($\RO$). We are particularly pleased that the reviewers acknowledge the unique challenges posed by the gradient inversion problem in the context of graph-structured data, specifically recognising that recovering the graph structure is a fundamentally different task compared to traditional gradient inversion problems ($\Rt, \RTH$). Furthermore, we appreciate the reviewers' recognition of the importance of the graph reconstruction metric introduced ($\RO, \RFI$), facilitating future research in this area. Finally, we are happy to read that the reviewers found our experiments to be  'extensive' and 'rigorous' ($\RF$), noting that GRAIN ‘outperforms existing baseline attacks' ($\RO,\RF$) and demonstrates 'promising performance across different scenarios' ($\Rt$). In the response below, we provide answers to common and important questions. We plan to incorporate their answers in the next revision of this paper. Further, we would like communicate to the reviewers that we are currently crafting responses for their outstanding questions not addressed in the main response and will make them available shortly.

---

> ### Author Response · Authors · 2024-11-22
>
> $\newcommand{\RO}{\textcolor{red}{61N5r}}$$\newcommand{\Rt}{\textcolor{blue}{d7Q7}}$$\newcommand{\RTH}{\textcolor{green}{HRGw}}$$\newcommand{\RF}{\textcolor{purple}{5Wij}}$$\newcommand{\RFI}{\textcolor{orange}{t4xV}}$$\newcommand{\grad}[1]{{\tfrac{\partial\mathcal{L}}{\partial #1}}}$$\def\colspan{{\text{ColSpan}}}$**Q.1 (Reviewers $\Rt, \RF$): What specific contributions does GRAIN make over DAGER (Petrov et al.)? What are the technical challenges specific to graph data and how does GRAIN overcome them?**
>
> We are grateful to the reviewers for the question, as it provides an opportunity to clarify and further detail the significance and contributions of our work.
>
> First, we want to emphasise that GRAIN is the first gradient inversion attack on GNNs. As such, an important contribution of the paper is to demonstrate that GNNs are indeed vulnerable to gradient leakage attacks. To reinforce this point, we draw the reviewer’s attention to the new results in **Q.5** of the Rebuttal, where we demonstrate that the uncovered vulnerabilities are general, working across dataset types (chemical and citations) and across architectures (GCN and GAT). Next, we outline some of the graph-data-specific challenges for gradient inversion attacks and how we tackle them.
>
> The first major challenge unique to GNNs is the introduction of the **unknown to the attacker** adjacency matrix $A$ that describes the graph structure of the input. In practice, this means that recovering the input features $\mathbf{X}$, usually a target of gradient inversion attacks, requires recovering the unknown graph structure and vice versa. Further, the matrix $A$ is often sparse and influences the gradient computation at multiple steps of the gradient computation (each GCN layer), making traditional gradient leakage much less effective in recovering it via optimisation. To tackle this, GRAIN makes two observations. First, the inputs of early GCN layers do not depend on the full graph structure but only on local degree-L neighbourhoods. Second, the theory developed by Petrov et al. [1], with the modification explained in the next paragraph, allows direct reconstruction of the inputs to those layers. With these, we are able to reconstruct the input $\mathbf{X}$ only based on local graph structures. However, this also means that unlike Petrov et al. [1], recovering the full graph structure, represented in terms of the adjacency matrix $A$, cannot be done based only on the gradients of the first few layers. To allow GRAIN to recover the full matrix $A$, and as additional filtering of wrong input features $\mathbf{X}$, we therefore developed our DFS-based traversal algorithm.
>
> The second major challenge is the introduction of the adjacency matrix $A$ in the gradients of $\grad{W}$. While the proof of Lemma 5.1 is heavily based on the results presented by Petrov et al. [1], its statement is, in our opinion, surprising. It states that individual input features to the GCN layers can be recovered from gradients **without knowledge of the structure of the graph**. Baseline attacks based on optimisation do not have this property, as the overall gradient of the network is heavily influenced by the exact matrix $A$. This is why, without knowledge of $A$, prior attacks achieve much worse results on node feature reconstruction. Further, as we discuss in **Q.2**, a graph-specific challenge to applying Theorem 5.1 from Petrov et al. [1] on graph gradients is the rank of $A$. In particular, in **Q.2**, we show an important generalization of the theory presented by Petrov et al. [1] that provides an exact condition for recovering individual input vectors $\mathbf{X}_i$ under any adjacency matrix $A$. Our experiments in Appendix B.2 in the latest revision of the paper, suggest that most input vectors $\mathbf{X}_i$ satisfy these conditions, explaining the efficiency of our filtering procedures for real-word graphs.
>
> Finally, an important contribution of our paper is achieving an efficient implementation of GRAIN on GPU. In particular, on top of an efficient GPU implementation of the spancheck of Petrov et al. [1], we also construct an efficient tensor representation of $L$-hop neighbourhoods that allows us to determine which blocks can be glued together in parallel, which is essential to achieving our practical results. Further, in **Q.5** we find that due to the sparsity of input features on some graphs, we can scale to much larger sets $\mathcal{T}^0$ compared to Petrov et al. [1], as we are able to recover individual features, avoiding the exponential explosion.
>
> We will include this discussion in the main paper in the next revision.

---

> ### Author Response · Authors · 2024-11-22
>
> $\newcommand{\RO}{\textcolor{red}{61N5}}$$\newcommand{\Rt}{\textcolor{blue}{d7Q7}}$$\newcommand{\RTH}{\textcolor{green}{HRGw}}$$\newcommand{\RF}{\textcolor{purple}{5Wij}}$$\newcommand{\RFI}{\textcolor{orange}{t4xV}}$$\newcommand{\grad}[1]{{\tfrac{\partial\mathcal{L}}{\partial #1}}}$$\def\colspan{{\text{ColSpan}}}$**Q.2 (Reviewer $\RFI$): If the adjacency matrix $A$ is rank-deficient, how does this affect the reconstruction capabilities of GRAIN? Are the adjacency matrices $A$ of real graph networks full rank?**
>
> We thank $\RFI$ for this great question! While it is the case that $A$ needs not to be full rank for real world graphs such as those in Tox21, most individual input vectors $\mathbf{X}_i$ to the GCN layers can still be recovered by GRAIN’s filtering procedure despite $A$ being rank deficient. We show this in the theorem below (Proof in Appendix A in the latest paper revision), which relaxes the full rankness condition of Lemma 5.1:
>
>  **Theorem**: Let there be a GCN layer with feature vectors $\mathbf{X} \in \mathbb{R}^{n\times d}$, a possibly-normalized adjacency matrix $A \in \mathbb{R}^{n\times n}$, and observed gradient update $\grad{W}\in \mathbb{R}^{d\times d}$ and $\mathbf{Z} = \mathbf{A}\mathbf{X}\mathbf{W}$. Assuming that both $\mathbf{X}$ and $\grad{Z}$ are full-rank, and $d < n$, then $\mathbf{X_i} \in \colspan(\grad{W})$ if and only if $A^T_i \notin \colspan(\bar{A_i})$, where $\bar{A_i}$ is the matrix $A$ with the $i$-th column removed.
>
> This means that in practice, GRAIN will be able to recover any input vector $\mathbf{X}_i$ for which its corresponding column in $A$ is linearly independent of the rest of the columns in $A$.
>
> Our practical experiments, both on synthetic graphs and graphs from the Tox21 dataset, shown in Appendix B.2 in the latest paper revision, demonstrate that while small graphs have adjacency matrices that are often low-rank, a very large percentage of the inputs to the first GCN layer can still be recovered under most circumstances. We also reaffirm the conclusions of [2] and [3], that real-world graphs are more often low-rank. We will include these results in the main paper for the next paper revision.
>
>
>
> **Q.3 (Reviwers $\RO, \RFI$):  What is the attack model of GRAIN? In particular, what are the capabilities and limitations of a potential adversary?**
>
> GRAIN is an honest-but-curious gradient inversion attack on Graph Neural Networks (GNNs) in Federated Learning (FL). In FL, multiple clients train a model locally and share weight updates with a server, which acts as the model aggregator. In GRAIN, the server is assumed to be an honest-but-curious adversary, aiming to recover training data solely through knowledge of the sent and received weight updates without interfering with the normal FL training protocol. In particular, the key assumptions of any honest-but-curious gradient inversion attacks, including GRAIN, are:
> - Clients truthfully report weight updates to the server.
> - The server adheres to the protocol without modifying model weights or architecture.
> - The server is knowledgeable of the input data structure, including the semantic meaning, value ranges, and normalization of individual input features.
> - The server has access to the original model sent to the client before the update.
>
>
> GRAIN does not assume knowledge of the client labels and targets the FedSGD protocol, where clients compute single-step weight updates. GRAIN is designed to be specifically applied to GNNs, and we present an implementation focused on Graph Convolutional Networks and Graph Attention Networks, for which we present state-of-the-art results in the paper and **Q.5**, respectively. To achieve this, we make the following additional assumptions:
> - The number of nodes in a graph is smaller than the embedding dimension.
> - All node features are discrete.
> - For any relevant linear layer (GCN layer $l < L$ in our experiments) $\mathbf{Z_l} = X_lW_l$, $\grad{\mathbf{Z_l}}$ is full-rank.
>
>
> In the main paper, we further assumed that the in-degree of each node is part of the feature vector and that the adjacency matrix $A$ is full-rank, such that Lemma 5.1 is applicable. We relax both of these assumption in **Q.6** and **Q.2**, respectively, showing they are not crucial for the operation of GRAIN.

---

> ### Author Response · Authors · 2024-11-22
>
> $\newcommand{\RO}{\textcolor{red}{61N5r}}$$\newcommand{\Rt}{\textcolor{blue}{d7Q7}}$$\newcommand{\RTH}{\textcolor{green}{HRGw}}$$\newcommand{\RF}{\textcolor{purple}{5Wij}}$$\newcommand{\RFI}{\textcolor{orange}{t4xV}}$$\newcommand{\grad}[1]{{\tfrac{\partial\mathcal{L}}{\partial #1}}}$$\def\colspan{{\text{ColSpan}}}$**Q.4 (Reviewer $\RF$): Why are the prior reconstruction quality metrics insufficient to measure the graph reconstruction quality? How did you ensure that the metrics you introduced are fair with respect to the baseline attacks?**
>
> We are grateful to reviewer $\RF$ for this question! We found it necessary to design our own set of metrics, as prior graph-related similarity measurements were not suitable for evaluating gradient inversion attacks. In particular, we wanted a metric that satisfies the following three qualities:
>
> - The metric should be efficiently computable in polynomial time
> - It should capture both structural and feature-wise information
> - Isomorphic graphs should be guaranteed to achieve a 100% score
>
> First of all, the NP-complete nature of the graph isomorphism problem makes it difficult to do any subgraph or full-graph matching, which we tackle by utilising the hidden states of a GCN to create an approximate matching between nodes.
>
> The second requirement is rarely satisfied by metrics defined in the literature (i.e. the edit distance), as comparison studies on coloured graphs are limited.
>
> Our solution to these problems was inspired by the ROUGE set of metrics, used for evaluation of textual similarity. Instead of comparing sequences such as unigrams or bigrams like ROUGE, we instead compute continuous properties of graphs on the scale of different k-hop neighbourhoods. This rationale allows us to compare both node-based and structural properties using a simple methodology.
>
> To ensure the fairness of the results of our experiments we selected two strong baselines – DLG as a staple multi-purpose gradient inversion attack, often applied in new gradient inversion domains, and Tableak, as it is an attack that has been optimised specifically for recovering a mix of continuous and discrete input features. Further, to account for the different nature of the baseline attacks and GRAIN, we systematically supply the baseline attacks with more information (including the correct number of nodes and for the **+A** variants of the baselines the full adjacency matrix), while penalising GRAIN as part of our metric when it fails to recover that same information. Still, GRAIN achieves noticeably better result, regardless, suggesting that the conclusions of the experiments are valid.
>
> Furthermore, prompted by reviewer $\RF$’s inquiry, we conducted a human evaluation study to measure the perceived reconstruction quality and compare it to our set of metrics. A group of 3 experts in Graph Theory and Chemistry were tasked to assign a reconstruction scores between 0 and 10 to each pair of prediction and client input on a mix of 120 samples from the Tox21, Clintox and BBBP datasets. Samples from both GRAIN and DLG were shuffled and anonymized before being presented to the participants. We then averaged the results and tabulated them in Table 6 in the Appendix of the latest paper revision. We observe very good correlation between our metrics and the reported human scores, even though our metrics are slightly more lenient to completely wrong reconstructions, compared to the evaluators. This leniency provides a slight advantage to the baselines when measured using our metrics, as the baselines fail catastrophically more often.

---

> ### Author Response · Authors · 2024-11-22
>
> $\newcommand{\RO}{\textcolor{red}{61N5r}}$$\newcommand{\Rt}{\textcolor{blue}{d7Q7}}$$\newcommand{\RTH}{\textcolor{green}{HRGw}}$$\newcommand{\RF}{\textcolor{purple}{5Wij}}$$\newcommand{\RFI}{\textcolor{orange}{t4xV}}$$\newcommand{\grad}[1]{{\tfrac{\partial\mathcal{L}}{\partial #1}}}$$\def\colspan{{\text{ColSpan}}}$**Q.5 (Reviewers $\RTH, \RFI$): Is GRAIN applicable to other architectures or datasets?**
>
> Yes! To showcase this, we apply GRAIN to a new architecture and a new dataset and show the effectiveness of the attack. Specifically, we explored the application of our work on Graph Attention Networks (GATs) and the Citeseer citation network dataset [4].
>
> First, we show that GATs can be attacked in an identical way to GCNs, as each node at every GAT layer is attended only by its neighbours. Therefore, the hidden state at the $l$-th layer is only determined by its $l$-hop neighbourhood, and can be filtered by the corresponding span check on the linear layer of the attention mechanism, similar to Petrov et al. [1]. We show that we achieve similar results to what we observed for GCNs, in particular a GRAPH-1 score of 90.7 on the Tox21 dataset for a GAT with a hidden dimension of $d^` = 200$ .
>
> GRAIN is similarly extendable to other datasets, such as citation networks. One additional challenge this type of data presents is the high dimensionality of $\mathcal{F}$, as these networks have binary features, corresponding to the appearance of a particular keyword in the paper/abstract. For instance, each node of the Citeseer dataset has 3,703 binary features, resulting in $\lvert \mathcal{F}\rvert=2^{3,703}$ different feature combinations. GRAIN can easily tackle this problem by recovering the features one-by-one by performing the span check on a row-wise truncated weight gradient. The remainder of the algorithm is trivially extendable. We applied GRAIN by utilising a heuristic search, that is described in **Q.7**, on the Citeseer dataset. We do so on subgraphs of similar sizes to the ones found in the molecular datasets, that we collect using multi-hop neighborhood sampling. We yield good initial results on the GAT architecture, with GRAPH-1 of 69.1.
>
> As so, we conclude that GRAIN’s scope covers both different models, and types of data. Any remaining numbers and details can be found in Table 10 in Appendix B.3 in the latest version of our paper.
>
>
>
> **Q.6 (Reviewers $\Rt, \RTH, \RFI$): Does GRAIN rely on node in-degree to be part of the feature vector, and does that limit the applicability of the attack?**
>
> The in-degree feature is not a requirement for GRAIN to work, but it does significantly lower its computational workload. In particular, it imposes restrictions during the exploration of building blocks during the filtering phase, and can help us determine an easier termination condition during the building phase.
>
> However, it is not necessary for GRAIN to use this information. Instead, in GRAIN we can simply generate a larger number of degree-1 building blocks before filtering them, and compute the gradient distance for each graph during the building. We apply this version of our attack on the Citeseer citation network dataset from **Q.5**, and achieve a GRAPH-1 score of 42.7, lower than the reported  69.1 with the in-degree feature, but also showing significant reconstruction capabilities. The full measurements are shown in Table 10 in Appendix B.3 of the latest paper revision.
>
> That said, we would like to reiterate that the node in-degree feature has been shown to provide the model with significant information about the graph structure, resulting in better accuracy, and, therefore, is part of many training real-world protocols. In many practical scenarios the attacker can leverage this information to reduce their computational complexity. All in all, we conclude that removing this feature is not sufficient for an effective defence, but can be a measure to increase the attacker’s computational load.

---

> ### Author Response · Authors · 2024-11-22
>
> $\newcommand{\RO}{\textcolor{red}{61N5r}}$$\newcommand{\Rt}{\textcolor{blue}{d7Q7}}$$\newcommand{\RTH}{\textcolor{green}{HRGw}}$$\newcommand{\RF}{\textcolor{purple}{5Wij}}$$\newcommand{\RFI}{\textcolor{orange}{t4xV}}$$\newcommand{\grad}[1]{{\tfrac{\partial\mathcal{L}}{\partial #1}}}$$\def\colspan{{\text{ColSpan}}}$**Q.7 (Reviewer $\RTH,\RF$): Can GRAIN leverage additional information in order to scale to larger graphs?**
> Yes! Similarly to prior optimization-based gradient leakage attacks in the image or text domains, additional prior knowledge about the particular graphs and their node features in the datasets can be incorporated in order to speed up the search for the correct building blocks at layer $l$ or to prioritize certain branches in the DFS search.
>
> In this context, we already employ a general ordering heuristic in our DFS algorithm to prefer building blocks with a lower distance score $S$, which allows us to begin the search from a subgraph that is very likely a part of the input. Next, we describe possible heuristics specific to different settings.
>
> For the citation networks considered in **Q.5** of the rebuttal, most nodes have a unique feature vector. As such, the tree search algorithm can be forced to prioritize paths that overlap nodes with identical feature vectors. Further, since each degree-2 building block is likely to be unique, we assign a lower preference score to blocks that are already part of the current graph, reducing the likelihood of a repeated selection as the algorithm progresses down the tree. This is more efficient than fully exploring the search space and results in us being able to recover a significant portion of the graph. An issue that we needed to address in this case is that large graphs often contain nodes with high in-degrees, for which exhaustively constructing all possible degree-1 neighbourhoods is expensive. To alleviate this, we first reconstruct as much of the graph as possible, and then exhaustively construct all possible graphs with edges between high-degree graphs. These heuristics allow us to recover the citation network with a score of GRAPH-1=69.1, compared to GRAPH-1=52.1 without them. The contribution to this improvement is most significant for graphs with a larger number of nodes ($\geq 25$), as we obtain a full reconstruction on 12/30 cases, compared to 1/30 without the heuristic. A full description of the resulting metrics can be found in Table 10 in Appendix B.3 of the latest revision.
>
> For chemical structures like those in Tox21, where these assumptions are unlikely to hold, we consider the opposite - we can assign a score to each graph, preferring ones with short cycles over ones with long ones, as many molecules contain arene rings with typical length of 5 and 6. Furthermore, a lower number of cycles are also preferable to a larger one, because the molecule will likely be more stable. As suggested by $\RF$, we can also use the chemical information given by the reconstructed features, such as the hybridization, which describes the bonds the atom participates in. We believe that such heuristics can be incredibly useful to lighten the computational complexity of our method, and that this is an interesting avenue for future work.
>
> [1] Petrov, Ivo, et al. "DAGER: Exact Gradient Inversion for Large Language Models." arXiv preprint arXiv:2405.15586 (2024).
>
> [2] Graph Structure Learning for Robust Graph Neural Networks. KDD 2020
>
> [3] Learning social infectivity in sparse low-rank networks using multi-dimensional hawkes processes. AAAI 2013.
>
> [4] Prithviraj Sen, Galileo Mark Namata, Mustafa Bilgic, Lise Getoor, Brian Gallagher, and Tina Eliassi-Rad. Collective Classification in Network Data. AI Magazine. 2008.

---

> > ### Author Response · Authors · 2024-11-29
> >
> > **Q.8 (Reviewer $\RTH,\RF$): Can GRAIN scale to large graphs ($\geq 25$ nodes)?**
> >
> > Yes, GRAIN can scale to larger graphs. We demonstrate this on the Pokec dataset [5], a social network dataset derived from the Slovakian social media platform of the same name. Most node features in the dataset, including eye color, body type, and hobbies, are categorical and can take many possible values. We one-hot encode them, while keeping the few remaining ordinal variables like age as continuous features. We sample 20 subgraphs for each of the size ranges 25-30, 30-40, 40-50, and 50-60 nodes to evaluate on. We demonstrate the results below:
> >
> >
> > ${\small
> > \begin{array}{r|cccc|c|}
> > n &   \text{GRAPH-0}    &   \text{GRAPH-1} &   \text{GRAPH-2} & \text{Full Reconstruction} & \text{Runtime [h]} \\\\
> > \hline
> > 25-30&98.3^{+0.2}\_{-0.4}&95.1^{+0.5}\_{-1.1}&96.8^{+0.4}\_{-0.9}&17/20&0.17 \\\\
> > 30-40&83.1^{+2.3}\_{-3.4}&61.6^{+3.1}\_{-3.0}&79.4^{+2.7}\_{-3.6}&5/20&0.46 \\\\
> > 40-50&69.3^{+3.2}\_{-3.8}&38.0^{+4.7}\_{-4.3}&59.2^{+3.7}\_{-4.0}&2/20&0.64 \\\\
> > 50-60&32.7^{+4.8}\_{-3.9}&23.3^{+4.2}\_{-3.5}&41.2^{+4.6}\_{-4.1}&3/20&0.43 \\\\
> > \hline
> > \text{Total}&70.9^{+6.2}\_{-6.5}&55.6\pm7.2&69.2^{+6.4}\_{-6.6}&27/80&1.70 \\\\
> > \hline
> > \end{array}}$
> >
> > We find that GRAIN equipped with the heuristics from **Q.7** is able to reconstruct much larger graphs in this setting, including some 60 node ones. Importantly, we find that our heuristic employing the feature-by-feature reconstruction of $\mathcal{T}\_0$ is more suited to the features of the Pokec dataset, allowing it to scale further, and that our tree search equipped with the prioritization of paths that overlap nodes with identical feature vectors has no trouble scaling to graphs of these sizes.
> >
> > [5] L. Takac, M. Zabovsky. Data Analysis in Public Social Networks, International Scientific Conference & International Workshop Present Day Trends of Innovations, May 2012 Lomza, Poland.

---

> ### Author Response · Authors · 2024-12-02
>
> We would like to thank the reviewers for their insightful comments, their crucial feedback, and for advising us on improving our paper. We believe that we have comprehensively addressed all of their concerns, provided new insights, and performed thorough experimental evaluations. As the deadline for the discussion is fast approaching, we would like to ask them to raise any outstanding concerns or give additional comments.

---

### Author Response · Authors · 2024-12-04

$\newcommand{\RO}{\textcolor{red}{61N5}}$$\newcommand{\Rt}{\textcolor{blue}{d7Q7}}$$\newcommand{\RTH}{\textcolor{green}{HRGw}}$$\newcommand{\RF}{\textcolor{purple}{5Wij}}$$\newcommand{\RFI}{\textcolor{orange}{t4xV}}$

We sincerely thank all five reviewers for their constructive comments and insightful questions, which have significantly helped us improve our work. We are particularly encouraged by the reviewers' recognition of our paper's strengths, as summarized below:

**Novelty**
- The problem is important ($\RTH$)
- The problem is unexplored ($\RO, \Rt, \RTH$)
- The problem is interesting and our paper can inspire further research in the area ($\RO$)
- Our paper is significant step toward understanding the privacy vulnerabilities of federated learning when applied to graph-structured data ($\RF$)
- GNN gradient inversion is fundamentally different task compared to traditional gradient inversion problems ($\Rt, \RTH$)
- The graph reconstruction metrics we introduced are important for future research in the area ($\RO, \RFI$)

**Extensive Experiments and Strong Results**
- The experiments are extensive and rigorous ($\RF$)
- GRAIN significantly outperform existing baseline attacks ($\RO,\RF$)
- GRAIN shows promising performance across different scenarios  ($\Rt$)

We acknowledge that our initial submission had areas for improvement. In response to the reviewers’ thorough reviews, we have provided the following additional information and experiments which we will incorporate in the next revision of the paper:

1. Better explanation of the contributions of our work, especially versus Petrov et al.

    - Clarified challenges specific to graph neural networks and how we solved them (**Q.1**)
    - Extended the theory introduced by Petrov et al. to handle GCN and GAT layers (**Q.1**)
    - Showed the role rank-deficiency of adjacency matrices plays in the reconstruction (**Q.5**)

2. Clarified the exact threat model assumed by GRAIN (**Q.3**), showing that in-degree features are not required for GRAIN to pose significant risks in practice (**Q.6**).
3. Provided an extended discussion regarding our novel graph reconstruction metrics (**Q.4**)

    - Added discussion on desired properties of the metrics
    - Added discussion on the motivation for our exact choice of metrics
    - Showed in a small user study that the metric results correlate well with human judgement

4. Showed that GRAIN is generic:

    - We showed that GRAIN is generic w.r.t. The GNN architectures it supports, showing that we can handle both GCNs (main paper) and GATs (**Q.5**)
    - We showed that GRAIN is generic w.r.t graph dataset types it supports, showing that it is applicable to chemical datasets (main paper), citation networks (**Q.5**), and social networks (**Q.8**)

5. Showed that GRAIN can scale:

    - In terms of number of input features (**Q.5**)
    - In terms of graph sizes (**Q.8**)
    - In terms of number of possible values per input feature (**Q1.7**)

6. Typos and paper clarifications:

    -  Provided a table summarizing all of our notations in the revised version of our paper (Table 5 in Appendix A).
    - Provided exhaustive clarifications to various technical questions the reviewers had, which we will incorporate in the next paper revision, alongside the reviewers’ other writing suggestions.

Once again, we deeply appreciate the valuable feedback and guidance provided by the reviewers.

Best regards,
The authors

---

### Meta-Review · Area_Chair_7xXB · 2024-12-20

**Metareview:**

This work introduces GRAIN, a gradient inversion attack tailored to Graph Convolutional Networks (GCNs) in federated learning settings. The primary contribution of this work lies in its tailoring these attacks to graph-structured data in federated learning, by devising a reconstruction attack capable of recovering both the graph structure and node features from shared gradient updates between clients and the server.

GRAIN employs a depth-first search strategy to identify subgraphs within gradients, iteratively filtering them by layer to reconstruct the full input graph. Notably, it leverages the low-rank structure of GCN layer updates and incorporates degree information in the features to generate the final prediction. Furthermore, the authors propose new evaluation metrics, such as GRAPH-N, to assess the similarity between reconstructed and original graphs in the context of graph gradient inversion attacks.

While the proposed method demonstrates promise, its effectiveness is limited when dealing with larger graphs, which is common for this type of attack. A more explicit connection to graph isomorphism would have strengthened the work, providing additional insights into the theoretical foundations of the task. Nevertheless, the reviewers' overall assessment was positive, and the paper has shown significant improvement following the rebuttal, which must make into the final version of the paper. Considering the borderline nature of this work, I recommend **acceptance if there is room**.

PS to authors: In one of the replies the authors said graph isomorphism is NP-complete. The paper also has a confusing statement: "Since exact matching of graphs is an NP-complete problem (Fortin, 1996)". The graph isomorphism problem belongs to NP but is not known to be either NP-complete or in P. In 2015, Babai found a quasipolynomial-time algorithm for graph isomorphism, running in exp((log n)^c) time for some constant c. If graph isomorphism were NP-complete, this would imply a collapse of the polynomial hierarchy. For the graph sizes considered in the paper, interactive color refinement works well and is reasonably fast.

**Additional Comments On Reviewer Discussion:**

The rebuttal was informative and addressed several key concerns raised during the review process. However, to further strengthen the paper, it would be beneficial for the authors to provide a more rigorous theoretical foundation for their approach. In particular, a deeper exploration of the underlying mathematical principles governing graph gradient inversion attacks would enhance the work's overall impact.

Notably, the authors' discussion should be improved by acknowledging the computational complexity of graph isomorphism, an aspect mistakenly described in the paper as "Since exact matching of graphs is an NP-complete problem (Fortin, 1996)" and in the rebuttal, which unfortunately was not caught by the reviewers during the initial evaluation. Adding a note or remark to clarify this point would not only demonstrate the authors' awareness but also provide valuable context for readers. We strongly expect this will be fixed in the paper.

---

### Decision · Program_Chairs · 2025-01-22

Accept (Poster)